# Deep mutation, insertion and deletion scanning across the Enterovirus A proteome reveals constraints shaping viral evolution

William Bakhache ®[1], Walker Symonds-Orr[1], Lauren McCormick ®[1,2] & Patrick T. Dolan ®[1]✉

Insertions and deletions (InDels) are essential to protein evolution. In RNA viruses, InDels contribute to the emergence of viruses with new phenotypes, including altered host engagement and tropism. However, the tolerance of viral proteins for InDels has not been extensively studied. Here, we conduct deep mutational scanning to map and quantify the mutational tolerance of a complete viral proteome to insertion, deletion and substitution. We engineered approximately 45,000 insertions, 6,000 deletions and 41,000 amino acid substitutions across the nearly 2,200 coding positions of the Enterovirus A71 proteome, quantifying their effects on viral fitness by population sequencing. The vast majority of InDels are lethal to the virus, tolerated at only a few hotspots. Some of these hotspots overlap with sites of host recognition and immune engagement, suggesting tolerance at these sites reflects the important role InDels have played in the past phenotypic diversification of Enterovirus A.

Genetic variation in RNA viruses arises through substitution, insertion and deletion. The influence of each of these mutational forces on the evolution of viral populations is determined by the rate at which they occur and their phenotypic impact, or fitness effect. Although single-nucleotide variants are the primary mechanism of adaptation to selective pressures on short evolutionary timescales, rarer insertions and deletions (InDels) can access evolutionary solutions inaccessible by substitution alone, sometimes dramatically altering protein structure and function. Both types of mutation have combined to set the tempo of the host–virus arms race and create the diversity of viruses and viral proteins we observe today. Here, we use emerging methods in mutational scanning to define the global mutational tolerance of a viral genome to substitutions and InDels.

Direct experimental measurement of mutational fitness effects (MFE) is now possible with deep mutational scanning (DMS), a transformative technology allowing measurement of the fitness effects of all possible non-synonymous, amino acid (AA)-changing mutations across cellular proteins[1,2]. In viral proteins, DMS studies have identified the constraints shaping mutational tolerance[3–8] and the mutational

pathways available to escape immune pressures[5,9]. However, conventional DMS approaches use PCR-based methods, which are largely limited to single-residue substitutions. Studies experimentally examining the effects of InDels on viral fitness have relied on high-fidelity population sequencing[10] or random transposon insertion mutagenesis[11,12]. Deep InDel scanning approaches that use synthetic oligo libraries encoding deleted or inserted peptide sequences have recently been developed to overcome these limitations[13–16]. Specifically, Saturated Programmable Insertion Engineering (SPINE) and Deep Indel Missense Programmable Library Engineering (DIMPLE)[14,15] have been used to explore InDel tolerance in cellular proteins.

Viruses in the order *Picornavirales* abound in our natural world, infecting a wide range of uni- and multicellular organisms. Picorna-like viruses are proposed to be ancestral to all modern viruses, imparting the core structural feature of icosahedral capsid proteins, the 'jelly roll' fold, and numerous modules involved in viral replication[17,18]. One genus of *Picornavirales*, Enterovirus, includes significant human pathogens that show a wide range of tissue tropism, modes of pathogenesis and immune profiles[19,20]. Of particular concern is the Enterovirus A (EV-A)

[1]Quantitative Virology and Evolution Unit, Laboratory of Viral Diseases, NIH–NIAID Division of Intramural Research, Bethesda, MD, USA. [2]Department of Biology, University of Oxford, Oxford, UK. ✉e-mail: patrick.dolan@nih.gov

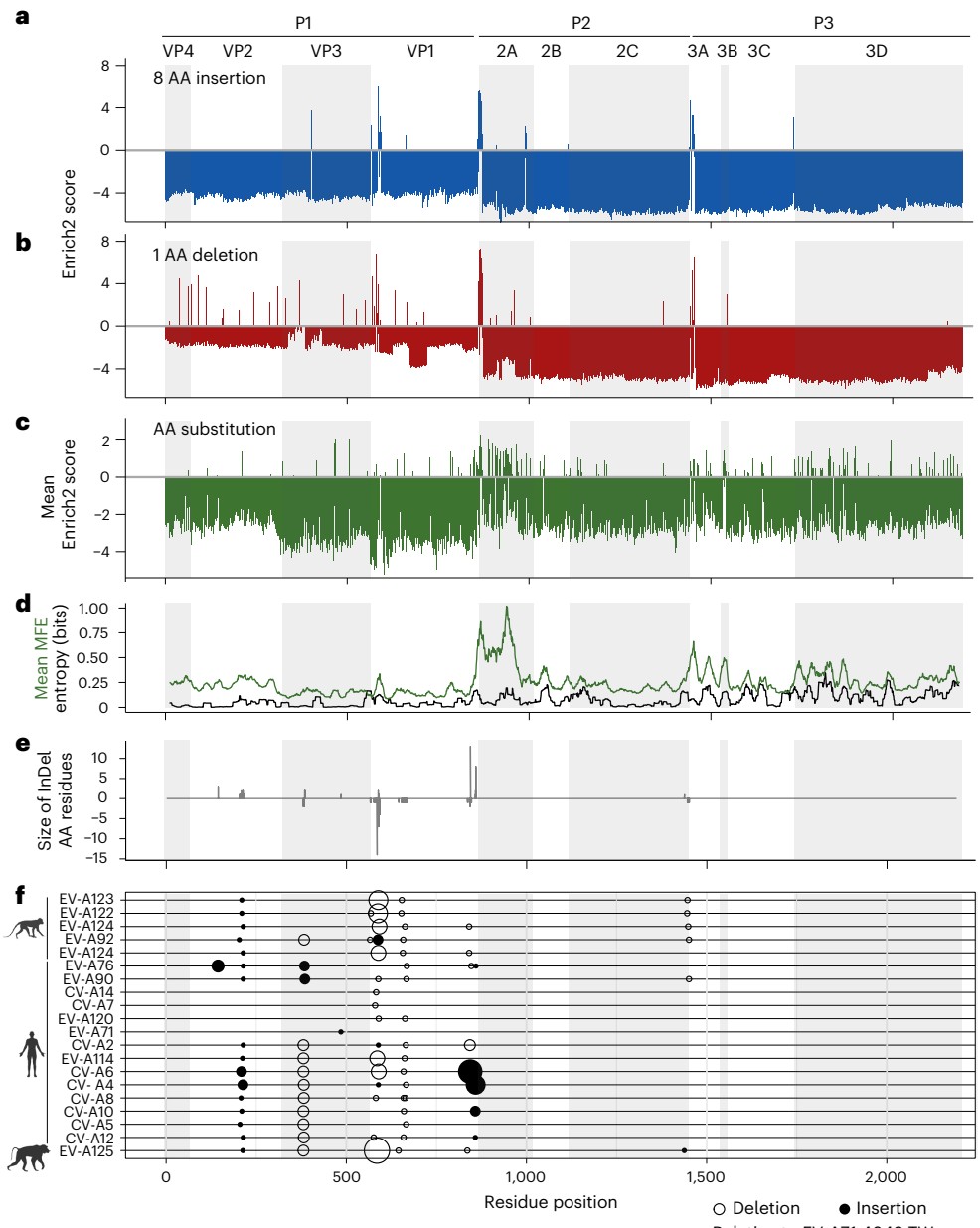

**Fig. 1 | Profiles of InDel and AA substitution tolerance in EV-A71. a–c**, Bar plot (bin = 1) showing the Enrich2 scores for 8 AA insertions (**a**), 1 AA deletions (**b**) and a median of all AA substitutions (**c**) for each coding position across the EV-A71 proteome. **d**, Line plot showing the sequence entropy from a balanced alignment of EV-A71 sequences and mean fitness effects for AA changes using a 21 AA sliding window across the EV-A71 proteome. **e**, Bar plot showing the size and position of specific InDels detected in other EV-A species relative to the EV-A71 4643 genome. **f**, The positions of individual insertions or deletions on the representative EV-A sequences shown in **e**. Alternating shaded regions highlight the boundaries between viral proteins in the viral polyprotein. TW, Taiwan. Illustrations of hosts in **f** created with BioRender.com.

species, which circulate worldwide and cause large outbreaks of hand, foot and mouth disease among young children[19]. Some genotypes, including EV-A71, can also cause severe, sometimes fatal, neurological complications including meningitis and acute flaccid myelitis[21,22]. EV-A71 has, therefore, been designated a prototype pathogen for understanding the biology, immunology and evolution of enteroviruses with pandemic potential[20].

## Results

### Global tolerance to insertion, deletion and substitution in EV-A71

To understand the comprehensive tolerance of the EV-A71 proteome to InDels and AA substitutions, we performed a large-scale mutational

screen across the complete EV-A71 coding sequence. The entire EV-A71 viral proteome is expressed as a single 2,193 residue polyprotein that is subsequently cleaved by viral proteases into the 11 viral proteins: the capsid proteins encoded in the P1 region (in genomic order: VP4, VP2, VP3 and VP1) and the replication proteins encoded in the P2 and P3 regions (2A, 2B, 2C, 3A, 3B, 3C and 3D), which have enzymatic and membrane remodelling activities[23,24]. Three separate strategies were used to generate mutational libraries across the viral polyprotein, described in detail in Methods. Briefly, we used two pipelines, SPINE and DIMPLE[14,15], to design and introduce oligonucleotide pools encoding either substitutions; deletions of 1, 2 or 3 codons; or a sequence known as an 'insertional handle' at each codon position into an infectious molecular clone of EV-A71. The insertional handle

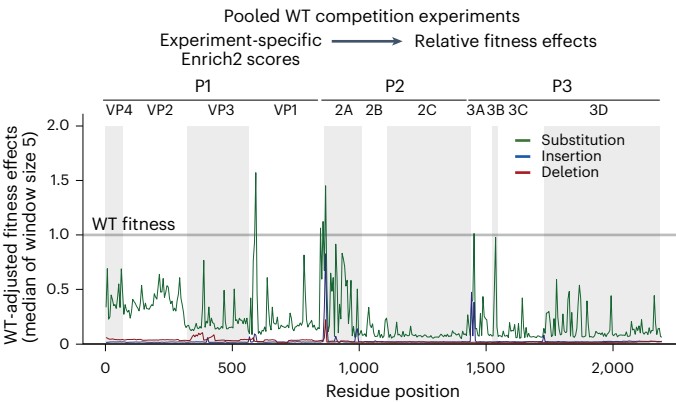

**Fig. 2 | Fitness effects of mutation from WT competition experiments.** Binned median relative fitness at 5 residue windows for 8 AA insertions, 1 AA deletions and all AA substitutions after adjusting according to pooled WT competition experiments (Supplementary Data 8). A fitness of 1.0 is equivalent to the parental WT reference strain. Alternating shading highlights the individual viral proteins in the EV-A71 polyprotein.

contains two outward-facing BsaI restriction enzyme sites that facilitate the subsequent cloning of any sequence of interest into each handle site, enabling rapid generation of libraries with a diverse range of inserts from the initial plasmid library. One caveat of this approach is that the BsaI sites used for the incorporation of the insert remain after insertion, leaving a fixed sequence flanking the inserted sequence of interest.

We generated initial virus library (passage 0) populations by transfecting in-vitro-transcribed viral RNA into rhabdomyosarcoma (RD) cells. Passage 0 virus populations were passaged at a low multiplicity of infection (≤0.1) to produce a passage 1 virus population. The passage 1 virus was used to infect cells for 9 h, after which total cellular RNA was collected. Mutagenized plasmid libraries and corresponding rescued and passaged viruses were sequenced either by long-read nanopore sequencing (for large insertions) or short-read sequencing (for small InDels and AA changes) (Extended Data Fig. 1a). The relative abundance of each variant in the population before and after selection was used to compute the enrichment using Enrich2, a widely used statistical framework for analysing DMS data[25]. The Enrich2 scores of the three independent biological replicates for all variant libraries were highly correlated, showing the reproducibility of our measurements (Extended Data Fig. 2).

Mapping patterns of constraint across the viral proteome, we identified regions of mutational tolerance (Fig. 1 and Extended Data Fig. 3). InDel mutations were tolerated at shared hotspot regions located at the amino termini (N termini) of VP1, 2A(pro) and 3A (Fig. 1a,b). In contrast, the viral proteome showed a broader tolerance to AA changes (Fig. 1c). In the capsid proteins, the N terminus of VP1 showed the highest tolerance to InDels, with residues 585 and 579 showing the highest scores for insertions and deletions, respectively. In the replication proteins, the N terminus of 2A(pro) showed notable tolerance to InDels, with residue position 863 having the highest score for insertions, and 865 for deletions. The highest enrichment for AA substitutions in the capsid proteins was at residue 662, present in the VP1 BC loop (L662T). The most enriched variant in the replication proteins was at residue 1,614, present in a loop in 3C(pro) (Q1614Y).

The distribution of MFE revealed that the vast majority of InDels were lethal to the virus, with only a small number tolerated (Extended Data Fig. 1b,c). This contrasted with the bimodal distribution observed for AA changes (Extended Data Fig. 1d), consistent with observations from experimentally passaged picornavirus populations[10,26]. We classified variant scores by enrichment and compared the proportion of variants in each class across the different viral proteins. In this

analysis, VP1, 2A(pro) and 3A appear most robust to InDels and AA changes (Extended Data Fig. 1e–g). The two largest replication proteins, 2C and 3D(pol), were the least tolerant to InDels.

Our experimentally measured scores for AA changes correlated with Shannon entropy measurements derived from natural variation across 482 complete sequences of EV-A71 (Fig. 1d). Notably, hotspots for InDels in the capsid proteins overlapped substantially with those occurring during the diversification of the EV-A species, a collection of 'serotypes' with distinct antigenic profiles and varying receptor usage (Fig. 1e,f). In contrast, the replication proteins showed little diversity of InDels among the EV-A species.

As our enrichment scores were computed independently for each experiment, comparing relative viral fitness between libraries required comparison to the reference genotype. We performed direct competition experiments by pooling one sublibrary from each of the capsid and replication protein libraries with the wild-type (WT) molecular clone plasmid and computing enrichment scores of each variant relative to the WT genotype (Extended Data Fig. 4). These normalized fitness estimates confirmed that most InDels are lethal to virus growth and only a small proportion of variants have neutral or beneficial fitness effects (Fig. 2). In fact, only a single 1 AA deletion at position 579 was more fit than WT (Extended Data Fig. 4g,h). Our WT-normalized scores and the enrichment scores from the complete EV-A71 proteome showed a good linear relationship (Extended Data Fig. 4i–n). Therefore, we were able to adjust the entire dataset relative to WT and place the scores on the same relative fitness scale to highlight the differences in tolerance to InDels and substitutions (Fig. 2).

## Content and context-specific influences on mutational tolerance

Through an initial analysis, we were able to interpret some general trends of mutational tolerance based on both mutational content and context. Our insertional handle strategy enabled us to generate a library of variable sequences at each insertion position, with each sequence consisting of a single variable AA flanked by flexible serine-glycine linkers. Analysis of the distribution of variants across enrichment bins showed differential tolerance for AAs. Proline, arginine, valine and phenylalanine were the least well tolerated, whereas lysine, tyrosine and asparagine showed the highest tolerance (Fig. 3a).

Sequence context-specific factors also contributed to differences in insertion tolerance. Insertion hotspots in VP3 and the N terminus of VP1 are more sequence specific, tolerating a select few AAs. Residues 402–410 in VP3 were highly tolerant to the AA tyrosine (Fig. 3b), whereas the N terminus of VP1 showed a preference for the AA lysine. In contrast, the N terminus of 2A(pro) showed a broader tolerance to most AAs. We also evaluated the effects of deletion length on tolerance, finding larger deletions tend to be more deleterious (Fig. 3c). VP1 and 2A(pro) tolerated larger deletions, whereas 3A showed greater sensitivity to deletion length (Fig. 3d). We also evaluated tolerance to the different AA residue substitutions (Fig. 3e). Similarly to the insertional library, proline was the least tolerated AA change, consistent with its ability to break secondary structures. Charged and aromatic AAs were also poorly tolerated. Better tolerated AAs were largely non-polar or polar neutral, with cysteine and methionine emerging as the best-tolerated AAs.

## Structural interpretations of EV-A mutational tolerance

We mapped the relative enrichment scores of InDels and AA changes on resolved structures of the EV-A71 capsid and replication proteins. 2A(pro) is a viral protease that cleaves the junction between the capsid and replication proteins. It also cleaves host proteins, contributing to the dampening of immune responses and the shutdown of host cap-dependent translation (Extended Data Fig. 5a,b)[27]. 2A(pro) features an active site with a catalytic triad (residues H21, D39 and a zinc-finger binding domain).

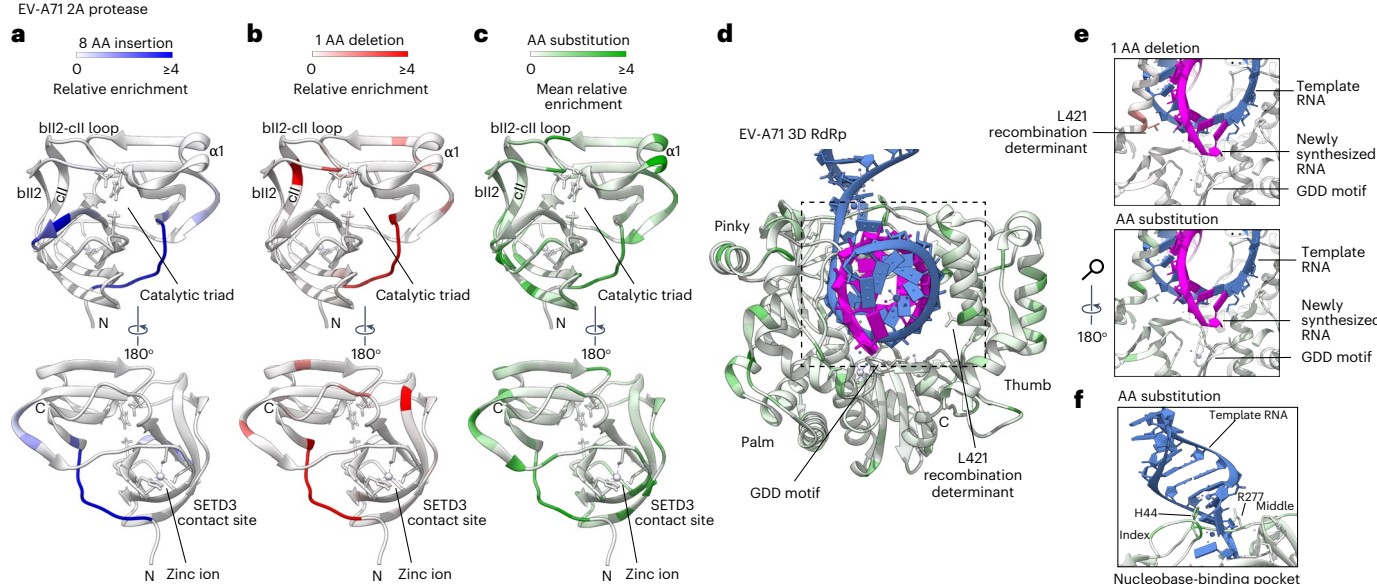

**Fig. 3 | Impact of altering insertion sequence, deletion length and AA residue substitution on EV-A71 growth. a**, Area plot showing the proportion of variants within Enrich2 score bins for all variable inserted residues. The two-sided chi-squared statistics are $\chi^2 = 108.69$ and d.f. = 38. **b**, Heat maps showing the Enrich2 scores in VP3 N termini VP1, 2A(pro) and 3A. The asterisk (*) represents the catalytic residue in the 2A(pro), H21. Black squares represent variants that were not analysed due to low variant counts in the input library. The diagram on the left explains the y axes of the maps. **c**, Area plot showing the proportion of variants within Enrich2 score bins for different deletion lengths. The two-sided chi-squared statistics are $\chi^2 = 63.252$ and d.f. = 4. **d**, Bar plots showing Enrich2 scores for different lengths at the N termini of VP1, 2A(pro) and 3A. Alternating shaded regions highlight the boundaries between viral proteins. **e**, Area plot showing the proportion of variants within Enrich2 score bins for all different AA substitutions. The two-sided chi-squared statistics are $\chi^2 = 991.35$ and d.f. = 38.

**Fig. 4 | Structural interpretation of InDel fitness effects for 2A(pro) and 3D(pol). a–c**, Tolerance for 8 AA insertion (**a**), 1 AA deletion (**b**) and substitutions (**c**) were mapped onto the structure of 2A(pro). **d**, AA substitution tolerance mapped onto the structure of the 3D(pol) elongation complex. **e**, Zoomed-in view of the interface between 3D(pol) and RNA, where scores for AA change and 1 AA deletion were mapped. **f**, Zoomed-in view of the nucleobase-binding pocket described for EV-A71, showing the side chains for the two main stabilizing residues. EV-A71 2A protease, PDB:3W95; EV-A71 3D RNA-dependent RNA polymerase (RdRp), PDB:6KWQ.

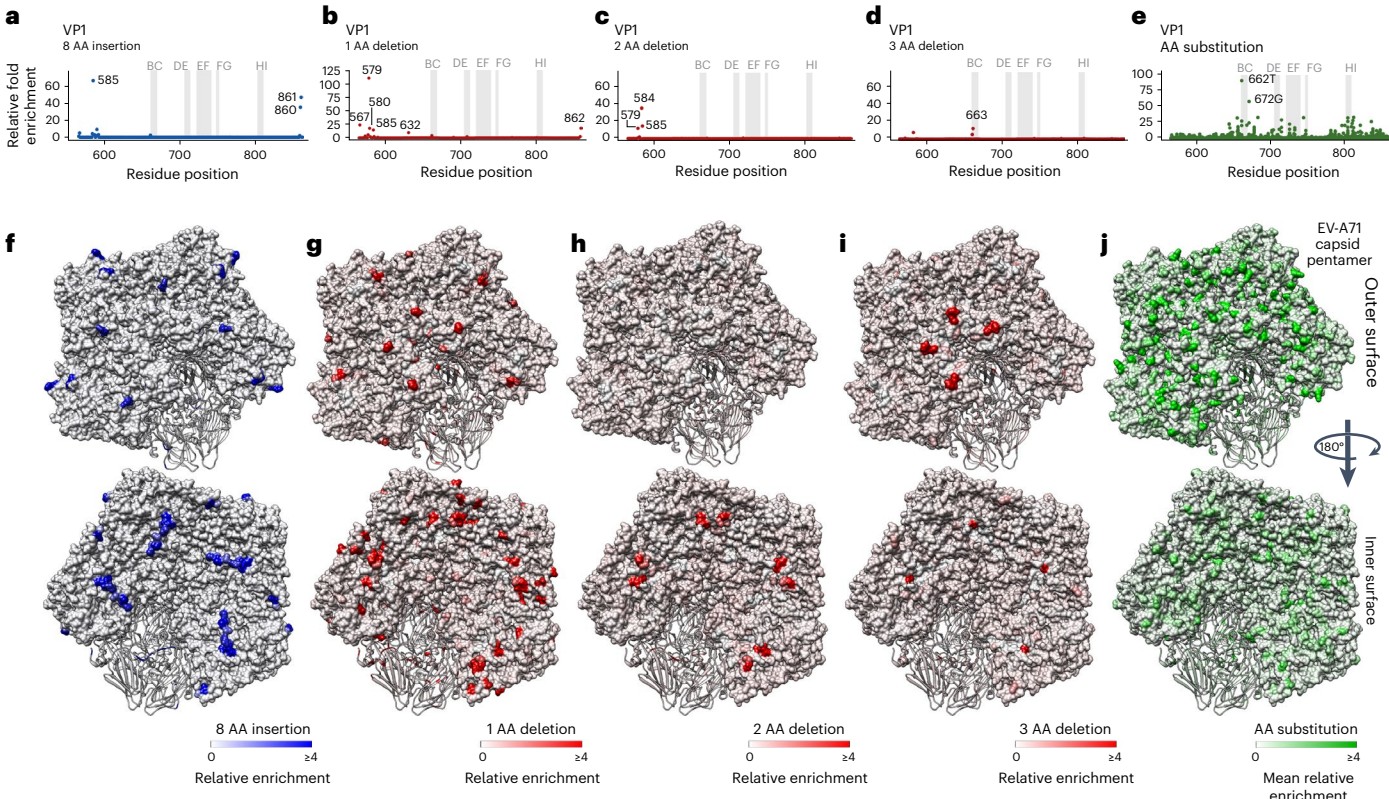

**Fig. 5 | Structural interpretation of mutational effects of the capsid.**
**a**–**e**, Manhattan plots showing the highly enriched variants for 8 AA insertion
(**a**), 1 AA deletion (**b**), 2 AA deletion (**c**), 3 AA deletion (**d**) and AA substitution (**e**)
across the VP1 protein. **f**–**j**, Relative enrichment values, that is, $2^{Enrich2}$, mapped on

the pentamer's external and internal surface for 8 AA insertion (**f**), 1 AA deletion
(**g**), 2 AA deletion (**h**), 3 AA deletion (**i**) and AA change (**j**). The PDB structure used
for the EV-A71 virion was PDB:8E2X.

We classified our variants in 2A(pro) according to secondary
structures[28], observing that loops were optimal InDel sites, whereas
helices were more tolerant to AA changes (Extended Data Fig. 5c).
Substitution with AAs with higher α-helical propensity was associated
with higher tolerance in α-helices, with alanine being the most well
tolerated (Extended Data Fig. 5d). Insertions (of eight residues) and
deletions of one residue were completely restricted from helices, prob-
ably due to altering the register of the helix and disrupting multiple
side chain interactions. Loops were also hotspots for InDels in other
replication proteins (3A and 2C; Extended Data Fig. 6a–d).

Notably, sites of constraint in 2A(pro) are consistent with known
host-facing functions. InDels and AA changes in regions proximal to
the active site and the zinc-finger binding domain were lethal to the
virus (Fig. 4a–c), confirming the importance of these sites for 2A(pro)
enzymatic activity. Similarly, the other viral protease, 3C(pro), did not
tolerate mutations at the active site (Extended Data Fig. 6e–g). The
interaction site of SETD3, a key host factor[29,30], on the surface of 2A(pro)
did not tolerate InDels. The N terminus, bII2-cII loop and α1-helix of
2A(pro) emerged as the most mutationally tolerant regions.

3D(pol), the RNA-dependent RNA polymerase (Extended Data
Fig. 5e,f), is the most conserved module of positive-strand RNA viruses[31].
InDels were rarely tolerated in EV-A71 3D(pol) (Extended Data Fig. 5g).
However, we observed tolerance to substitutions in helices and loops.
Again, AAs with higher α-helical propensity were better tolerated in
α-helices (Extended Data Fig. 5h). We mapped tolerance onto the EV-A71
3D(pol) elongation complex[32] to understand how contacts with template
RNA affect mutational tolerance. Interestingly, a residue that determines
recombination rate, L421 (ref. 33), showed tolerance for a 1 AA dele-
tion, suggesting minimal fitness costs in this context (Fig. 4d). The GDD
motif (residues 328–330), essential for polymerase activity, showed no

tolerance. A nucleobase-binding pocket formed by the side chains of
residues 44 and 277 in 3D(pol) stabilizes template RNA interactions in
EV-A71 (ref. 32) (Fig. 4d). We found that the threonine residue at position
44 was robust to AA changes, with higher scores for aromatic AAs and
histidine (Extended Data Fig. 5i), whereas R277 was mutationally con-
strained, consistent with the important role it plays in RNA stabilization[32].

**Evolutionary constraint in the EV-A71 capsid**
An infectious EV-A71 virion comprises 60 subunits of capsid protomers,
each consisting of 4 proteins (VP1, VP2, VP3 and VP4) that enclose a viral
genome[34]. InDels were predominantly enriched at the N and carboxy (C)
termini of VP1 (Fig. 5a–d). In general, we noted tolerance to both substi-
tutions and InDels in surface-exposed regions (Fig. 5). Remarkably, the
BC loop displays distinct tolerance, accommodating an 8 AA insertion
and deletions of 1 AA and 3 AA in length. This is consistent with observa-
tions in poliovirus, an EV-C, in which the BC loop also shows remarkable
mutational tolerance to insertions[35]. The BC loop also showed tolerance
to AA substitutions, with the variant L662T (VP1 Thr97) most well toler-
ated (Fig. 5e). The inner surface, where the N terminus of VP1 is located,
is highly enriched for InDels (Fig. 5a,b). In contrast, the external surface
of the pentamer had higher enrichment for AA substitutions compared
with the inner surface, in agreement with recent phylogenetic analysis
of Enterovirus capsids[36] (Fig. 5e,j). The surface-exposed VP1 loops
(Extended Data Fig. 7e) contain several neutralizing epitopes, underscor-
ing the importance of mutational robustness in this region of the capsid.

**InDel-tolerant sites overlap with points of phenotypic
divergence**
To understand the evolutionary history of the InDel-tolerant sites iden-
tified in our scan, we generated a phylogenetic tree from a multiple

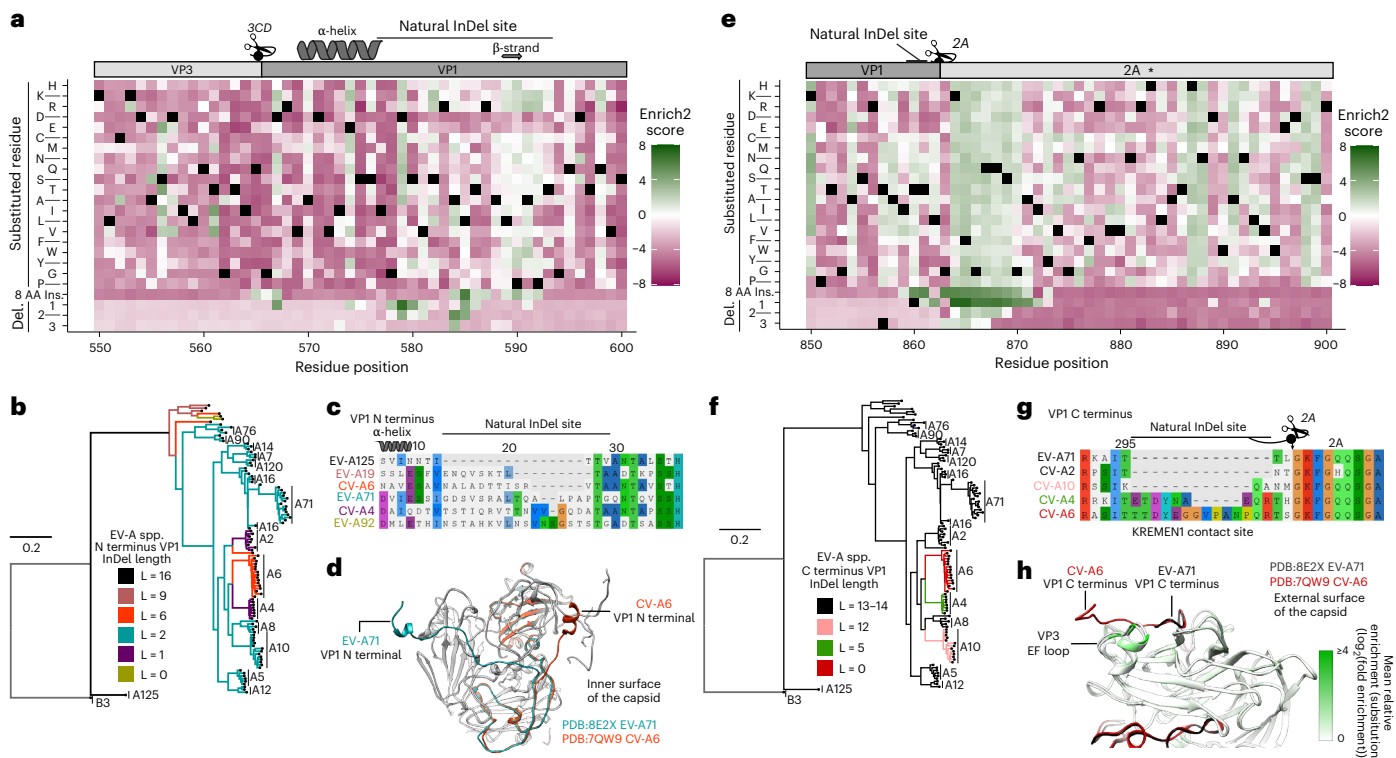

**Fig. 6 | InDels as a contributor of EV-A species evolution. a,e,** Heat map plotting the Enrich2 scores for mutations at the InDel hotspot regions present at the N (**a**) and C (**e**) termini of VP1. Phylogenetic tree based on the full coding sequences of EV-A species (*n* = 107 sequences). **b,f,** The phylogenetic tree is coloured by the gap lengths observed at the N (**b**) and C (**f**) termini of VP1. **c,g,** Multiple sequence alignments of the N (**c**) and C (**g**) termini of VP1. **d,h,** Structural alignment of EV-A71 (PDB:8E2X) and CV-A6 (PDB:7QW9) pentamer at the N (**d**) and C (**h**) termini. Mean relative enrichment scores for AA changes were mapped onto

the EV-A71 and CV-A6 structures in **h**. The asterisk (*) in the heat map represents the residue H21 present in the catalytic triad of the 2A(pro). The cleavage sites where 3CD cleaves VP3 and VP1 and the 2A(pro) cleaves the capsid proteins from the replication proteins are shown. Black squares for AA changes are the WT sequence. Black squares for deletions are variants removed due to low variant counts in the input library. L is equal to the length of the different gaps. Del., deletion; ins., insertion.

sequence alignment of representative EV-A proteome sequences (*n* = 107) (Fig. 6b,f). This analysis revealed a complex evolutionary history at the N terminus of VP1 (Fig. 6b,c). EV-A125, basal on the tree, had the longest gap length. On the distal branches, insertions have been accumulating, with a gap length of 2 residues being the most prevalent. Interestingly, a switch from gap length 2 to shorter or longer gaps has occurred multiple times, a pattern unique to this site. CV-A6 (coxsackievirus A6) and EV-A71 VP1 N termini appeared as linear epitopes at the inner surface of the capsid (Fig. 6d). InDels were highly enriched at that site, and a cluster of residues tolerated AA changes with a preference for positively charged AAs (Fig. 6a). The evolutionary history of InDels at the C terminus of VP1 followed a simpler pattern. Most EV-A species had a gap length of 12–14, including non-human enteroviruses and most circulating human enteroviruses (Fig. 6f). An insertion event occurred before the emergence of CV-A4 and CV-A6 (Fig. 6f,g). The insertion in the CV-A6 VP1 C terminus led to the extension of the C-terminal end, which comes into contact with the receptor kringle containing transmembrane protein 1 (KREMEN1) and covers the EF loop of VP3 (ref. 37) (Fig. 6h). The VP3 EF loop was highly enriched for AA changes, suggesting immune pressures act on this region. These data highlight the unique roles of InDels in shaping the structural plasticity of VP1 and EV-A diversification.

## Discussion

InDels play an important role in the evolution of RNA viruses[31,38,39]. However, experimental exploration of evolutionary landscapes in RNA viruses has been primarily focused on non-synonymous mutations. In this work, we generated the most comprehensive

map yet of MFE in a viral proteome, including over 45,000 insertions, 6,000 deletions and 41,000 AA substitutions (Extended Data Fig. 3). The resulting data reveal new insights into the understudied role of InDels, and their relationship to AA substitutions, in the context of viral protein evolution[12,40,41]. Our data highlight the deleterious load that InDels place on viral populations, with >95% of InDels characterized as lethal to the virus. Tolerant sites were found in several hotspot regions, consistent with recent comprehensive deep sequencing of InDel diversity in poliovirus-infected and dengue-infected cells[10].

One key observation emerging from our study is the complex constraints governing InDel and substitution tolerance in the EV-A71 capsid. The capsid proteins thread a difficult evolutionary needle, maintaining an extracellular and metastable state that protects the genome in harsh conditions while also being labile enough to rapidly uncoat after engaging host receptors or host environments. They are also the target of selection, usually carrying key immune epitopes. In our screen, VP1 had the greatest concentration of positions with high relative tolerance to all mutation classes, specifically at the N and C termini (Fig. 5a–e), consistent with our analysis of natural variation across EV-A (Figs. 1d–f and 6). These host- and viral-facing functions overlap in picornaviral capsid proteins; however, in enveloped viruses such as flaviviruses and coronaviruses, these functions are largely partitioned between their spike or envelope proteins and their nucleocapsid proteins, which appear to relax evolutionary constraints on these proteins[10,42]. Consistent with this, studies examining the effects of mutations, including single-codon deletions, have shown much greater tolerance to deletions in the spike protein of coronaviruses

than what is observed here, largely mirroring where substitutions were tolerated[43,44].

The consequences of variations in VP1 may reflect important historical phenotypic shifts. For example, we observed insertions in the C terminus of VP1 in CV-A4 and CV-A6 relative to other EV-A genotypes (Fig. 6f). On the basis of recent structures of the EV-A71 and CV-A6 virions, this insertion extended the VP1 C terminus to form a contact site with KREMEN1, which serves as a receptor for CV-A4, CV-A6 and several other EV-A genotypes. The extension of the VP1 C terminus also covered the VP3 EF loop, a mutationally robust region in our AA substitution screen (Fig. 6h). This suggests that immunogenic properties and host engagement might be linked through pleiotropy and that selection driven by immune pressure could drive diversification to new tropism.

Similarly, the N terminus of VP1 has accumulated multiple InDels along the evolutionary descent of EV-A but little variation is observed within EV-A71 isolates (Figs. 1f and 6b). The mature virion of enteroviruses has been shown to undergo transient conformational changes termed 'viral breathing', where the major structural rearrangement is the reversible exposure of the VP1 N terminus to the external surface[45,46]. During the virus uncoating, pH and entry host factors trigger the irreversible externalization of the VP1 N terminus, leading to viral genome release[47–49]. This region has also been described as highly immunogenic[50–53]. Together, these observations highlight the critical host- and virus-facing functions of Enterovirus VP1, placing the VP1 N terminus at the centre of Enterovirus entry and immune engagement and resulting in its evolution under conflicting selective pressures. We speculate that past selection has shaped the VP1 N terminus into a highly evolvable, structurally flexible region of the genome that can tolerate this selective landscape, evolve to overcome immune bottlenecks and explore new host ranges through altered host factor engagement[39].

Information on the mutational constraint-shaping InDels in viral population diversity is critical to understand the mutational pathways available to viruses to evade immune recognition but could also inform vaccine engineering. As a specific example, switching of surface-exposed loops in VP1 has recently been shown as a potential approach to engineer vaccines against EV-A71 and CV-A16 (ref. 54). Our data would be valuable in developing candidates for such approaches. More broadly, the comprehensive nature of these studies reveals engineering principles that could be used to train synthetic biology technologies towards better vaccine and immunogen design.

Finally, from an evolutionary perspective, these data describing the constraints shaping substitution and InDel diversity in Enterovirus populations only provide half the picture. In the future, it will be important to connect these data with complementary work measuring mutation rates in enteroviruses[10,26], investigating the mechanisms of mutation generation[33] and describing their evolutionary dynamics globally[55] to develop a complete understanding of evolutionary potential in enteroviruses and understand how insertion, deletion and substitution converge with selection to create the diversity we observe in modern enteroviruses.

## Methods

### Cells and reagents

RD cells (CCL-136; ATCC) used in infection experiments were maintained at 37 °C with 5% $CO_2$ cultured in Dulbecco's Modified Eagle Medium (30-2002; ATCC) supplemented with 10% foetal bovine serum (FBS) (10437-028; Gibco). Infection experiments were performed under the same conditions, except the concentration of FBS was reduced to 5%.

### Generation of domesticated EV-A71 Tainan/4643/98 molecular clone

As SPINE and DIMPLE use BsaI and BsmBI type IIS restriction enzymes to assemble mutant libraries, we first 'domesticated' the original EV-A71

strain Tainan/4643/98 molecular clone (GenBank: AF304458.1)[56] by removing 10 BsmBI or BsaI sites to improve efficiency of downstream assembly steps. Two contiguous regions of the molecular clone did not contain BsaI or BsmBI restriction sites; these were subcloned by PCR. Four fragments encoding the rest of the plasmid were generated synthetically by Twist Biosciences, removing natural BsaI and BsmBI sites by replacing them with synonymous codons. Synonymous mutations at codons within the restriction sites were used to remove them. All the fragments (6 in total) had a 30 bp overlap and were assembled by NEBuilder HiFi DNA Assembly (E5520S; New England Biolabs). Whole plasmid sequencing was performed by Plasmidsaurus using Oxford Nanopore Technology with custom analysis and annotation. We compared virus production by 50% tissue culture infectious dose ($TCID_{50}$) assay between the original and domesticated clones and observed that viral growth in RD cells was comparable with rescued virus titre reaching ~$10^6$ $TCID_{50}$ ml$^{-1}$. Sequences for the fragments used to build this clone, and the full-length molecular clone sequence, can be found in Supplementary Data 1.

### Mutagenesis library design

Three separate strategies were used to generate mutational libraries. We used previously reported pipelines, SPINE and DIMPLE[14,15], to computationally divide the polyprotein coding sequence of the EV-A71 Tainan/4643/98 molecular clone into 'sublibrary fragments' (Supplementary Fig. 1a). Deletion libraries, encoding deletions of 1, 2 and 3 codons across the entire polyprotein open reading frame, were directly encoded in an oligonucleotide pool of 45 fragment sublibraries. Each sublibrary was amplified and assembled into a plasmid backbone fragment lacking the cognate region. Missense variants, in which every possible AA change was introduced at all positions in the coding sequence, were encoded in the oligonucleotide pool as 42 sublibrary fragment assemblies. Insertion libraries were generated by directly encoding an in-frame 8 residue peptide sequence (SGRPGSLS), known as an insertional handle, at each codon position in the oligonucleotide pool as 28 sublibrary fragment assemblies. The insertional handle contains two outward-facing BsaI restriction enzyme sites that facilitate the subsequent cloning of any sequence of interest into each handle site, enabling rapid generation of libraries with a diverse range of inserts from the initial plasmid library.

We designed separate insertion, deletion and AA substitution libraries for both the capsid (that is, the P1 protein) and the replication proteins (that is, the P2–P3 proteins), for a total of six libraries. To design capsid and replication protein insertion libraries, SPINE[14] was used. A FASTA file containing the sequence of the domesticated EV-A71 molecular clone was provided where the first nucleotide of the first and last codon of capsid (746 and 3,331) and replication proteins (3,332 and 7,324) were specified.

The following command was used to run the SPINE code for insertion libraries:

```
> python3 run_spine.py -wDir inputdirectory
-geneFile bbfree.fasta -oligoLen 300 -mutationType
DIS
```

We used the DIMPLE[15] GUI interface to design deletion and AA substitution libraries, with start and end nucleotide positions (for P1, 743 through 3,328; for P2–P3, 3,329 through 7,321) for capsid and replication proteins, respectively. The software output included a list of oligos in each oligo pool, representing diversified sublibrary fragments, and a list of primers for inverse PCR and for oligo pool PCR. These sequences are available in Supplementary Data 2–3, 4–5 and 6–7 for insertion, deletion and AA substitution libraries, respectively, with sequences listed in FASTA format and sections demarcated by a label row beginning in '#'. Oligo pools were ordered from Twist Biosciences. Primers for Deletion_Capsid_15 were manually redesigned because

this reaction produced an off-target truncated amplicon instead of the desired full-length amplicon; the redesigned forward and reverse primers had the sequences ATACGTCTCcccggagcccccaagccag and ATACGTCTCgtacccattcgggttggttgtgccttc, respectively, where overhang nucleotides are capitalized.

## Mutagenesis library construction

Mutagenesis libraries were constructed as outlined in ref. 15. Briefly, to construct each sublibrary, oligos containing the diversity of interest were amplified from the respective oligo pool. Inverse PCR was used to amplify the remainder of the molecular clone, and Golden Gate cloning was used to ligate the fixed backbone amplicon to the diversified insert sequence. Q5 High-Fidelity DNA Polymerase (M0491L; New England Biolabs) was used to generate backbone amplicons, followed by Dpn1 treatment (R0176L; New England Biolabs) to remove template plasmid and gel purification (T1020L; New England Biolabs). Oligo pool amplification was performed using KAPA HiFi HotStart PCR Kit (KK2502; Roche). To verify the presence of a single amplicon of the correct length, products were analysed on an Agilent TapeStation 4200 using High Sensitivity D1000 ScreenTape (5067-5584; Agilent). To produce a mutagenized sublibrary, 300 ng inverse PCR product was mixed with 20 ng corresponding oligo pool PCR product in an NEBridge BsmBI-v2 Golden Gate assembly reaction (E1602L; New England Biolabs) with 60 cycles of 5 min digestion at 42 °C and 5 min ligation at 16 °C. Ligation products were cleaned up (T1030L; New England Biolabs) and transformed in NEB 10-beta Electrocompetent *E. coli* (High Efficiency) (C3020K; New England Biolabs) according to the manufacturer's recommendations. A small amount of outgrowth was plated to check for adequate variant coverage, defined as ≥100 cfu times the number of designed variants in the sublibrary. Transformed cells were grown in a 50 ml LB media culture containing 100 µg ml⁻¹ of carbenicillin (J67159.AE; Thermo Fisher Scientific) at 37 °C for 14 h and then purified using the QIAGEN Plasmid Midiprep Kit (12145; QIAGEN). Sublibraries corresponding to each mutagenesis library were pooled in equimolar amounts to constitute the complete libraries.

## Generation of 5 AA insertion library

The insertion libraries generated in the previous step contained an insertional handle, enabling downstream replacement with any sequence of interest. To remove any contaminating domesticated molecular clone (used as template for inverse PCR reactions), a chloramphenicol cassette was designed, flanked with inward-facing BsaI sequences to replace the insertional handle and outward-facing BsmBI recognition sequences to replace the antibiotic resistance cassette with a sequence of interest; this sequence is available in Supplementary Data 8. To do this, 300 ng of the capsid or replication protein insertional handle library was mixed with 20 ng of the chloramphenicol cassette and cloned using the NEBridge BsaI-v2 Golden Gate Assembly Kit (E1601L; New England Biolabs) with 30 cycles of 5 min digestion at 37 °C and 5 min ligation at 16 °C. Ligation products were cleaned up and transformed in NEB 10-beta Electrocompetent *E. coli* (C3020K; New England BioLabs) according to manufacturer recommendations. Transformed cells were grown in a 50 ml LB media culture containing 100 µg ml⁻¹ of carbenicillin and 25 µg ml⁻¹ of chloramphenicol at 37 °C for 14 h and then purified using the QIAGEN Plasmid Midiprep Kit. A small amount of outgrowth was plated to check for adequate variant coverage, defined as ≥100 cfu times the number of designed variants in the library. To construct 5 AA insertion libraries, a BsmBI-flanked oligo pool with the insertion GS-X-SG was used (Twist Bioscience; see Supplementary Data 9). X represents every AA, such that this oligo pool contained 20 variants in total. The 5 AA insertion libraries were assembled as described in the previous step from the chloramphenicol insertion library and the 5 AA oligo pool, except with the NEBridge BsmBI-v2 Golden Gate Assembly Kit.

## Generation of mutational libraries at VP1 and 2A(pro) N termini with a WT plasmid

Mutational sublibraries at the N termini of VP1 and 2A(pro) and the original EV-A71 molecular clone were pooled in equimolar amounts. The sublibraries used for the N terminus of VP1 were sublibrary 8 for insertion, sublibrary 13 for deletion and sublibrary 12 for AA change. For the N terminus of 2A(pro), the sublibrary 1 of all the different mutational libraries was used. WT count was estimated by measuring the synonymous change present at residue position 642 when comparing the original (codon: GAG) and 'domesticated' (codon: GAA) molecular clones.

## Generation of virus libraries

Plasmid libraries and viral molecular clones were linearized downstream of the poly(A) tail using the enzyme EagI-HF (R3505L; New England Biolabs) at 37 °C overnight. Linearized plasmid was cleaned up using the Monarch PCR & DNA Cleanup Kit and used as a template for the HiScribe T7 High Yield RNA Synthesis Kit (E2040S; New England Biolabs). In-vitro-transcribed viral RNA was cleaned up (T2040L; New England Biolabs) and transfected into RD cells using the TransIT-mRNA Transfection Kit (MIR 2250; Mirus Bio) using 0.5× the recommended RNA and reagent concentrations. After 2 days of transfection, cells were subjected to 2 freeze–thaw cycles. To remove cellular debris, the supernatant was centrifuged at 2,000*g* for 5 min. Virus rescue efficiency was evaluated by titrating the supernatant using $TCID_{50}$ (ref. 57). Titre for the EV-A71 molecular clone was ~$10^6$ $TCID_{50}$ ml⁻¹.

For insertional handle libraries, 3.25 µg in-vitro-transcribed RNA ($\sim 7.6 \times 10^{11}$ molecules) was transfected into 2 million cells to generate passage 0 virus. For 5 AA insertion, deletion and AA change libraries, 9.85 µg in-vitro-transcribed RNA ($\sim 2.3 \times 10^{12}$ molecules) was transfected into 6 million cells. Library rescue efficiency varied between $10^3$ and $10^4$ $TCID_{50}$ ml⁻¹ for insertional handle libraries, $10^3$ and $10^4$ $TCID_{50}$ ml⁻¹ for 5 AA insertion libraries, $10^5$ and $10^6$ $TCID_{50}$ ml⁻¹ for deletion libraries and ~$10^6$ $TCID_{50}$ ml⁻¹ for AA change libraries.

To generate passage 1 virus, RD cells were washed once with PBS (30-2200; ATCC) and then incubated with low inoculum (multiplicity of infection ≤ 0.1) of passage 0 virus for 1 h at 37 °C. Then, media was added to the cells and infection was allowed to continue for 24 h before collecting the virus via freeze–thaw as in the generation of passage 0 virus.

## Sequencing insertion, deletion and mutational scanning libraries

Infections were performed with passage 1 virus as described in the previous section but they were allowed to proceed for 9 h after addition of media for intracellular RNA extraction and subsequent sequencing. Intracellular RNA extraction was performed using the QIAGEN RNeasy Kit (74106; QIAGEN). To generate cDNA for sequencing, ProtoScript II First Strand cDNA Synthesis Kit (E6560L; New England Biolabs) was used to generate first-strand cDNA using reverse primers for P1 or P2–P3. First-strand cDNA was used as a template in 4 independent Q5 PCR reactions with 25 cycles. Primers for the capsid region were TCAAATTCATTTTGACCCTCAACACA (forward) and TAGATAGCTCCGGACTGCTGTC (reverse); primers for the replication protein region were TCAAAGCCAACCCAAATTATGCT (forward) and TGGTTATAACAAATTTACCCCCACCA (reverse). Primers for amplifying a region covering the N termini of VP1 and 2A(pro) were GCAATCGTCTGTCACCCTTGTA (forward) and CAATCCCCTGGTTCCGAATGAC (reverse). The input plasmid library was prepared for sequencing as above, beginning with the PCR step. Amplicons were gel purified before sequencing library preparation. Nanopore sequencing was performed for the insertional handle experiments. Sequencing libraries were prepared using the Native Barcoding Kit 24 V12 or V14 (SQK-NBD112.24 or SQK-NBD114.24; Oxford Nanopore Technologies) according to the manufacturer's instructions and were sequenced on a MinION device. Illumina sequencing libraries were prepared using the Twist Biosciences

Enzymatic Fragmentation 2.0 Kit with Universal Adapters (104207; Twist Biosciences) with 180–220 bp target fragment sizes. Illumina libraries for 5 AA InDel experiments were sequenced individually with the MiSeq reagent kit v.2, 300 cycle (MS-102-2002; Illumina). Illumina libraries for all replicates for both capsid and replication protein AA change libraries were pooled and sequenced with a NextSeq 2000 P3 flow cell, 300 cycle kit (20040561; Illumina). The mutational library focused at the N termini of VP1 and 2A(pro) with a WT plasmid were pooled and sequenced with a NextSeq 2000 P1 flow cell, 600 cycle kit (20075294; Illumina).

### Sequence analysis
Nanopore sequencing reads were basecalled using the high-accuracy module of the neural network basecaller guppy (guppy_gpu/6.0.6 or guppy/6.5.7), producing FASTQ files from FAST5 or POD5 files for each sample. bcl2fastq (bcl2fastq/2.20) was used to demultiplex Illumina sequencing reads. All sequencing reads were mapped using minimap2 (minimap2/2.24 or minimap2/2.26) with the -ax flag set to map-ont or sr for Nanopore or Illumina reads, respectively. For insertion and deletion libraries, mapped sequencing reads were processed using custom scripts stickleback.py and smelt.py, respectively. Mutational scanning libraries were mapped with the GATK/Analyze Saturation Mutagenesis tool[58]. To remove codon counts attributable to sequencing error, we used a custom script codonFilter.r. We then removed WT codon counts and converted reads to hgvs format for use in Enrich2.

### Enrich2 analysis
To assess changes in variant frequency after selection (viral passage), we used Enrich2 with the scoring method set to Log Ratios (Enrich2) and normalization method set to Library Size (All Reads)[25]. For the mutational libraries at the N termini of VP1 and 2A(pro) with a WT plasmid, the WT normalization method was used. Due to variant count drop-off in the input deletion and 5 AA insertion libraries, a minimum variant counts threshold was applied. The minimum count for capsid and replication deletion libraries was set to 50 and 20, respectively. The minimum count for 5 AA insertion libraries was set to 1.

### Structural analysis
The Protein Data Bank (PDB) structures used for the structural mapping were PDB:3W95 (EV-A71 2A), PDB:5GQ1 (EV-A71 2C), PDB:6HLW (EV-A71 3A), PDB:3OSY (EV-A71 3C), PDB:3N6L or PDB:6KWQ (EV-A71 3D), PDB:8E2X (EV-A71 virion) and PDB:7QW9 (CV-A6 virion). Secondary structure assignment was performed using the 2Struc web server[59] using the STRIDE assignment method. For mapping relative enrichment of variants onto the structures, an 'attribute' text file compatible with UCSF Chimera (v.1.16) was generated, modifying the positions to align with the structural information of the PDB file. Within Tools/Structural Analysis, the 'define attribute' and 'render by attribute' functions were used to apply the 'attribute' file to a given structure and colour the structure by the relative enrichment values. The 'Find Contacts' function in the default settings was used to determine residues in EV-A71 3D (PDB:6KWQ) that are in contact with the template RNA.

### Phylogenetic and entropy analysis
All complete EV-A protein sequences were downloaded from NCBI Virus and then clustered at 98% by cd-hit, yielding 107 sequence clusters. The representative sequences derived from this clustering were then aligned by MAFFT. Next, the aligned sequences were indexed to the EV-A71 strain Tainan/4643/98 and the starting positions of all gaps were recorded. A second MAFFT alignment was performed using the same 107 EV-A clusters using the ICTV Enterovirus B exemplar isolate sequence (GenBank:AAB59927.1) as an outgroup. A phylogenetic tree was produced from this alignment using the maximum-likelihood method in RAxML with 100 bootstrap replicates. The PROTGAMMAWAG option was selected in RAxML, which is for AAs with a Γ rate

heterogeneity model using a WAG substitution matrix. FigTree was used for phylogenetic tree visualization and customization. AliView was used for visualization and representation of the multiple sequence alignment.

To assess protein sequence diversity within EV-A71, a set of 482 sequences was aligned and Shannon entropy ($H(x)$) at each residue was calculated, with gaps counted as a separate character: $H(x) = -\sum_{i=1}^{21} p_i \log_2 p_i$, where $i$ is each AA (or gap character) and $p_i$ is the number of times that character is observed divided by the total number of sequences. A 21 AA rolling window mean was calculated, starting at position 10 and ending at position 2,185, and plotted.

### Statistics, reproducibility and data analysis
Statistical data analysis and visualization was performed with R (4.3.0) and Graphpad Prism 10. R packages used for figure generation include ggplot2, tidyverse, tidyr, ggpubr, dplyr, ggridges, ineq, RColorBrewer, stringr, gglorenz, readr, scales, zoo, Biostrings and DescTools. Base R code was used for $\chi^2$ and linear model analysis (lm function). Mutational scanning experiments shown in Figs. 1 and 2 are the means of three independent biological replicates. The 5 AA insertion scanning experiments shown in Fig. 3 are the means of two independent biological replicates.

### Biological materials
All biological materials, plasmids, cell lines and libraries are available by request to the corresponding author. Reasonable requests will be granted within 2 months, or as soon as possible, pending any necessary material transfer agreements or other restrictions.

### Reporting summary
Further information on research design is available in the Nature Portfolio Reporting Summary linked to this article.

## Data availability
All processed data used to generate the figures and analysis reported here are included in an accompanying Dryad repository (https://doi.org/10.5061/dryad.866t1g1xm)[60]. All raw sequencing read data are publicly available in the NCBI Short Read Archive under project number PRJNA1066851. The protein structures used for the structural mapping are available in the Worldwide Protein Data Bank (WWPDB.org), under the following PDB identifiers: PDB:3W95 (EV-A71 2A), PDB:5GQ1 (EV-A71 2C), PDB:6HLW (EV-A71 3A), PDB:3OSY (EV-A71 3C), PDB:3N6L or PDB:6KWQ (EV-A71 3D), PDB:8E2X (EV-A71 virion) and PDB:7QW9 (CV-A6 virion). Source data are provided with this paper.

## Code availability
Python scripts used in mapping the reads with engineered insertions and deletions are available at https://github.com/QVEU/InDel_Toolkit. The R scripts associated with specific bioinformatic computations performed herein are available at https://github.com/QVEU/eva71_dimple. Rscripts to regenerate all the figures are included in the Dryad repository associated with this article, along with all the python and R script versions used for this study: https://doi.org/10.5061/dryad.866t1g1xm (ref. 60).

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

## Acknowledgements

This work was supported by funding to P.T.D. from the Division of Intramural Research, NIH–NIAID, under project number 1ZIAAI001360. The funders had no role in study design, data collection and analysis, decision to publish or preparation of the manuscript. We acknowledge the foundational work done in this area by E. Ehrenfeld and colleagues in the Division of Intramural Research, NIH–NIAID which inspired this work. We acknowledge the help of W. Coyote-Maestas, J. S. Fraser and D. Nedrud from the University of California San Francisco with the SPINE and DIMPLE pipeline. We thank the laboratory member B. A. Catching for his assistance with structural interpretations. We acknowledge J. Lack, team lead of the Collaborative Bioinformatics Resource at the NIAID, for help with uploading sequencing data to the SRA database. L.M. is a DPhil student in the NIH Oxford–Cambridge Scholars Program.

## Author contributions

P.T.D. and W.B. conceived this study. W.B. and W.S.-O. performed the experiments and collected data. L.M. curated phylogenetic and sequence data. W.B., P.T.D., W.S.-O. and L.M. performed the analysis and composed the article.

## Competing interests

The authors declare no competing interests.

## Additional information

**Extended data** is available for this paper at https://doi.org/10.1038/s41564-024-01871-y.

**Correspondence and requests for materials** should be addressed to Patrick T. Dolan.

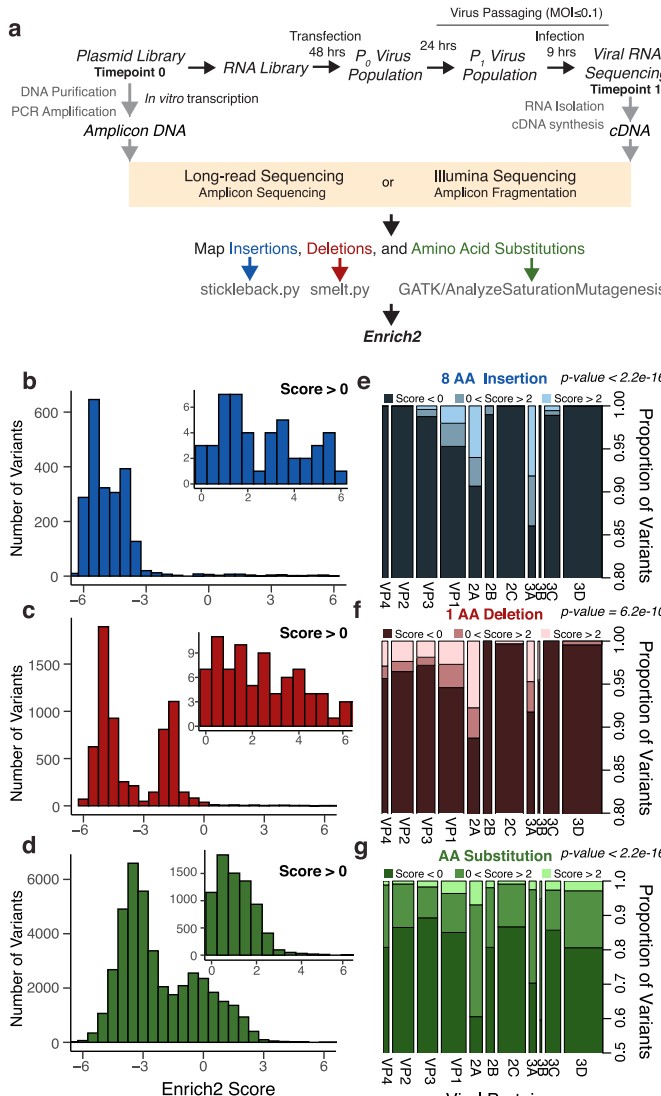

**Extended Data Fig. 1 | Distribution of Enrich2 scores and protein specific classification of tolerance. (a)** Workflow of the experimental virology pipeline to rescue virus from plasmid libraries and the subsequent sequencing steps at different timepoints to measure variant frequency change in the population. The technologies used for sequencing and the analysis pipelines to detect different variants are detailed. **(b–d)** Histograms visualizing the distribution of variant scores were shown for **(b)** 8 AA insertion, **(c)** 1 AA deletion, and **(d)** AA substitution; inset histograms show the distribution for variants with a score higher than 0. Area plot showing the proportion of variants within Enrich2 score bins for 8 AA insertion **(e)**, 1 AA deletion **(f)**, and AA change **(g)** across different viral proteins. The width of the column represents the total number of variants for each protein. The two-sided chi-squared statistics for **(e)** df = 20, $\chi^2$ = 140.43, **(f)** df = 20, $\chi^2$ = 84.685, and **(g)** df = 20, chi-squared = 1793.8.

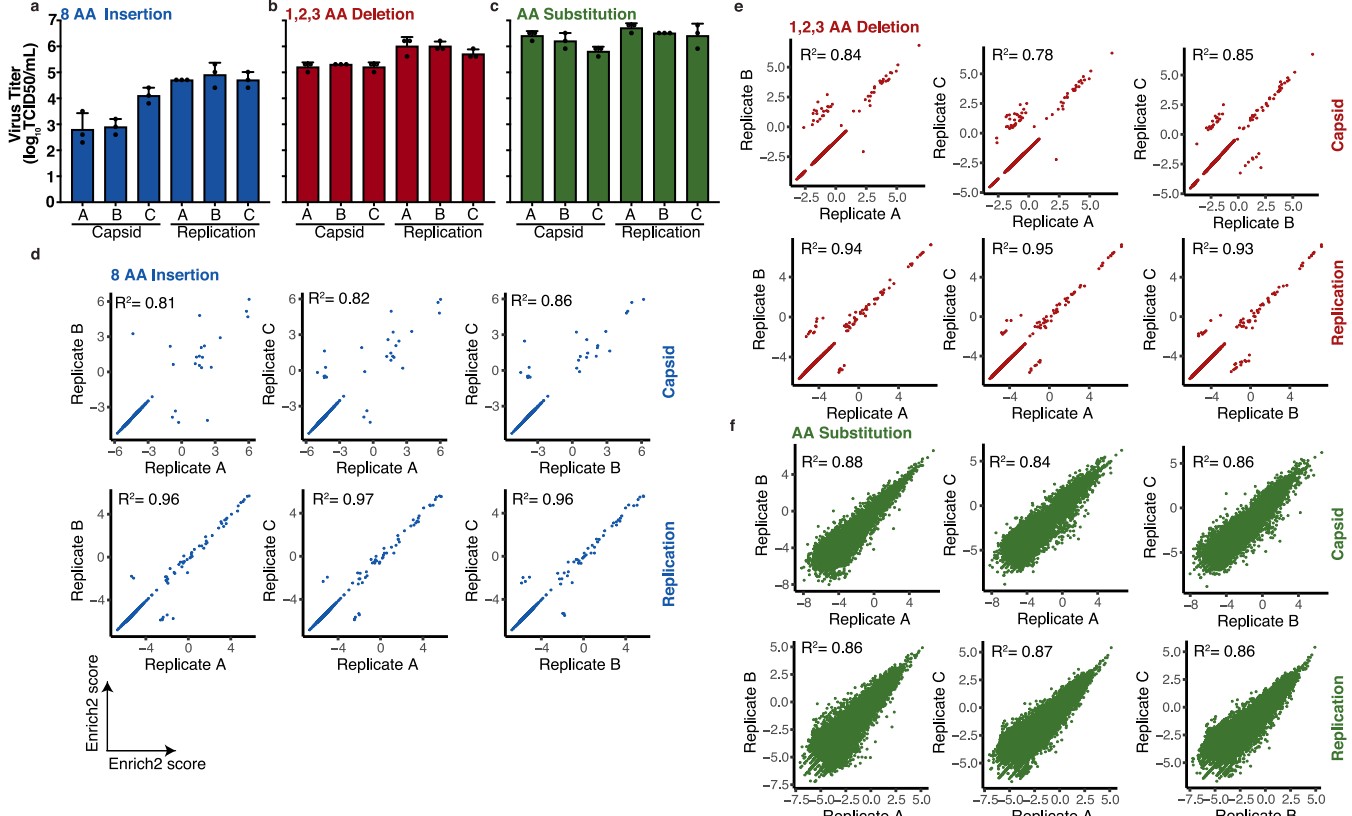

**Extended Data Fig. 2 | Reproducibility of the Deep Insertion, Deletion, and Mutational Scanning experiments.** (**a**–**c**) Bar graphs showing passage 0 virus rescue efficiency (TCID50/mL) for insertion (**a**), deletion (**b**), and AA change (**d**) libraries for the three biological replicates. Data are presented as mean values ± SD. Scatter plots showing the Enrich2 scores of the different biological replicates for insertion (**d**), deletion (**e**), and AA change (**f**) libraries.

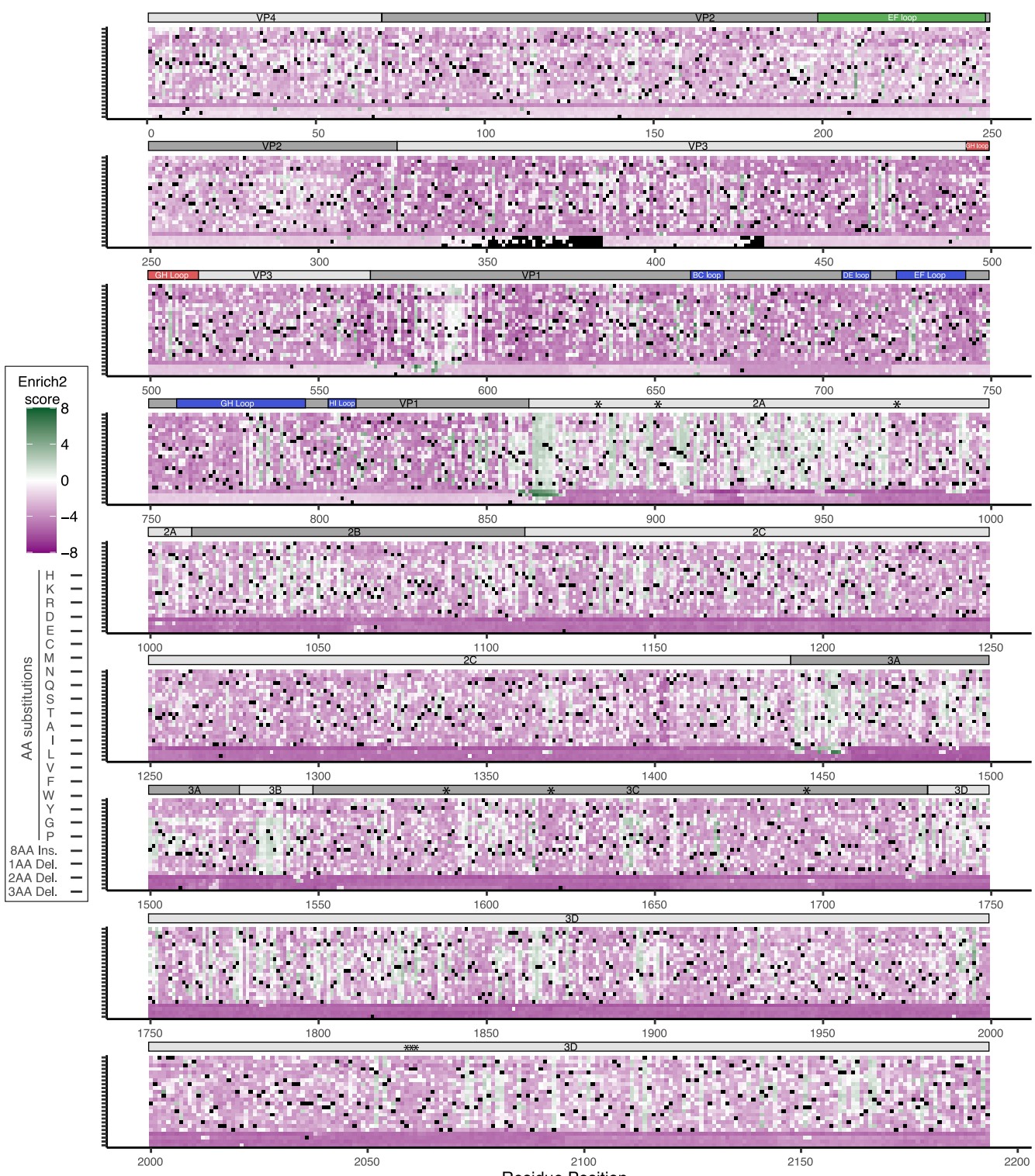

**Extended Data Fig. 3 | Enrich2 scores for insertions, deletions, and AA changes across the EV-A71 proteome.** Heatmap showing the Enrich2 score of mutations across the EV-A71 proteome. The different surface-exposed loops in the capsid proteins were annotated with rectangles of different colors (VP2: green, VP3: red, VP1: blue). Stars represent active sites of the different replication proteins. In 2A(pro) and 3C(pro), these stars highlight the catalytic triad. For 3D(pol), the stars represent the GDD motif. Black squares for AA changes are the WT sequence. Black squares for deletions are variants that were removed due to low variant counts in the plasmid library.

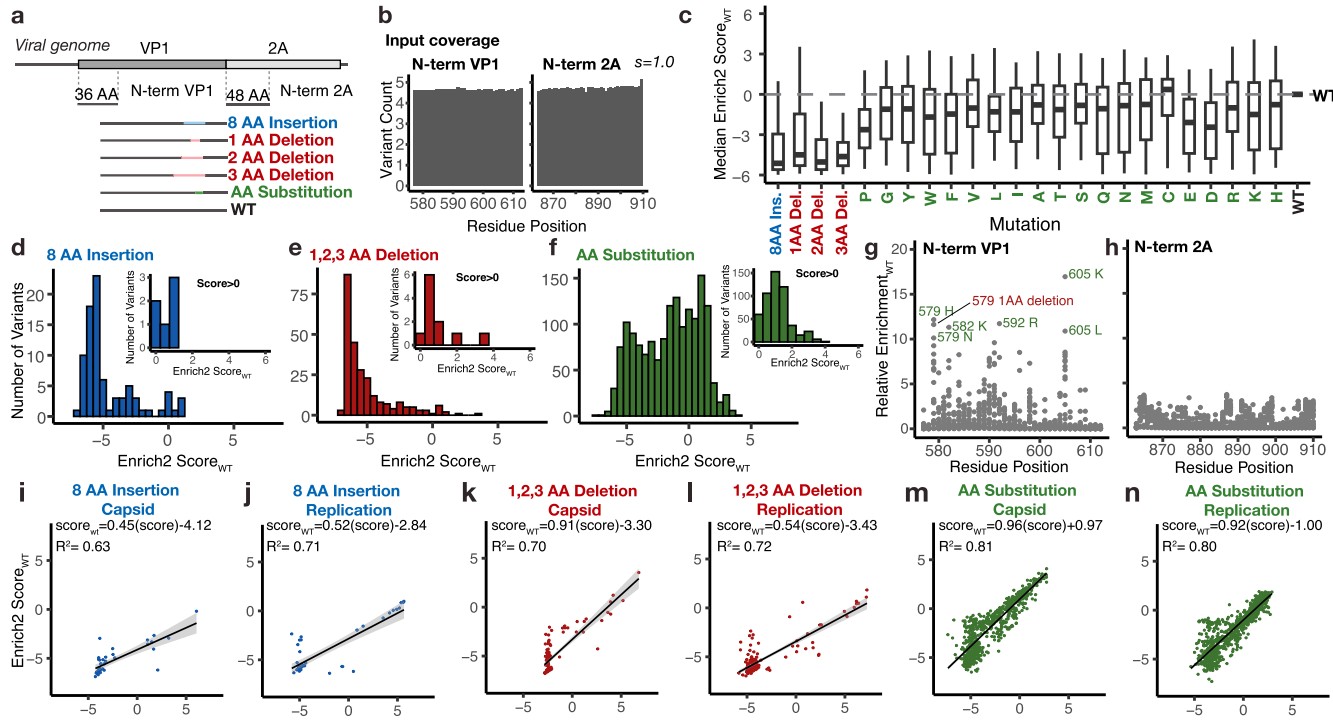

**Extended Data Fig. 4 | Competition experiments using pools from InDel hotspot regions.** (**a**) Visual representation of the target regions selected for the generation of a mutational library containing insertions, deletions, AA changes, and a WT plasmid. (**b**) Bar plot (bin = 1) showing the variant counts for the input plasmid library at the N-termini of VP1 and 2A(pro). All mutation types are stacked. (**c**) Box plot displaying the median Enrich2 scores for the different variants in the library. The lower hinge of the box plot represents the 25th percentiles and the upper hinge represents the 75th percentiles, with the center line showing the median. The whiskers extend to the smallest and largest values within 1.5 times the interquartile range (IQR) from the hinges. The dashed gray line represents the Enrich2 score for the WT sequence, set at 0. Histogram showing the distribution of fitness effects for AA insertion (**d**), deletion (**e**), and substitution (**f**). Inset histograms highlight variants with enrichment scores higher than wildtype. Manhattan plots displaying the enriched variants relative to WT at the N-termini of VP1 (**g**) and 2A(pro) (**h**). (**i**–**n**) Scatter plots of the Enrich2 scores from the complete mutational screen (x-axis) plotted against the WT normalized Enrich2 scores derived from the targeted mutational screen (y-axis) for insertions (**i,j**), deletions (**k,l**), and substitutions (**m,n**) in the capsid (**i,k,m**) and non-structural (**j,l,n**) regions of the genome. The grey shade area represents the 95% confidence interval. A linear model was used to derive a formula estimating the WT normalized score for all the complete mutational libraries.

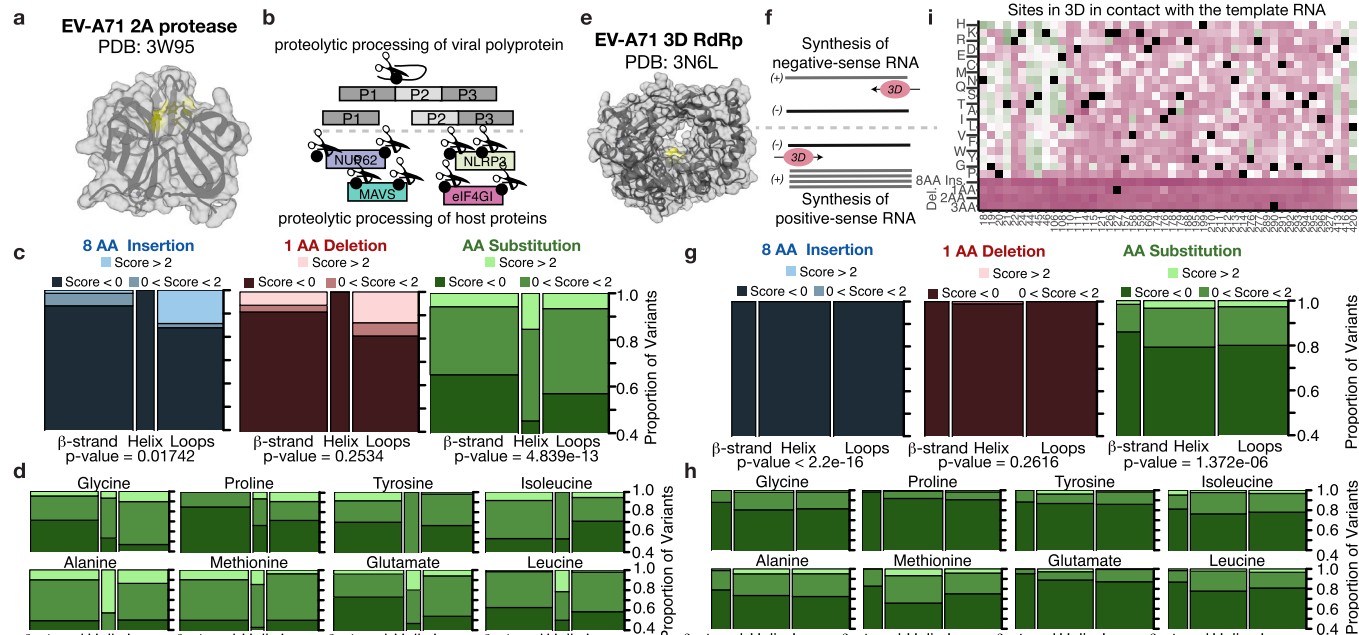

**Extended Data Fig. 5 | Structural interpretation of InDel fitness effects for 2A(pro) and 3D(pol).** (**a**) Structure of the EV-A71 2A(pro) (PDB: 3W95) with active site in yellow. (**b**) Cartoons showing the proteolytic activity of the 2A(pro) in processing the viral polyprotein and host proteins. (**c**) Area plot showing variants classified by Enrich2 score across the different secondary structure assignments in the 2A(pro). The width of the column represents the total number of residues that adopt a given secondary structure. The two-sided $\chi^2$ statistics in panel (**c**) for 8 AA insertion $\chi^2 = 11.99$ df = 4, 1 AA deletion $\chi^2 = 5.3477$ df = 4, and AA change $\chi^2 = 63.698$ df = 4; for 8 AA insertion: $\chi^2 = 73.753$ df = 2, 1 AA deletion $\chi^2 = 2.6817$ df = 2; and AA change: $\chi^2 = 32.706$ df = 4. (**d**) Area plots showing the proportion of substitutions within Enrich2 classes for different secondary structures in the 2A(pro). (**e**) The structure of the EV-A71 3D(pol) with the GDD motif highlighted

in yellow (PDB: 3N6L). (**f**) Cartoon showing the role of 3D(pol) in replication of viral genomes. (**g**) Area plot showing variants classified by Enrich2 score across the different secondary structure assignments in the 3D(pol). The width of the column represents the total number of residues that adopt a given secondary structure. The two-sided $\chi^2$ statistics in panel (**g**) for 8 AA insertion $\chi^2 = 73.753$ df = 2, 1 AA deletion $\chi^2 = 2.6817$ df = 2, and AA change $\chi^2 = 32.706$ df = 4. (**h**) Area plots showing the proportion of substitutions within Enrich2 classes for different secondary structures in the 3D(pol). (**i**) Heatmap showing the Enrich2 scores of the 3D(pol) contact residues with the template RNA. Black squares for AA substitutions represent the WT sequence. Black squares for deletions represent variants that were removed due to low variant counts in the input library.

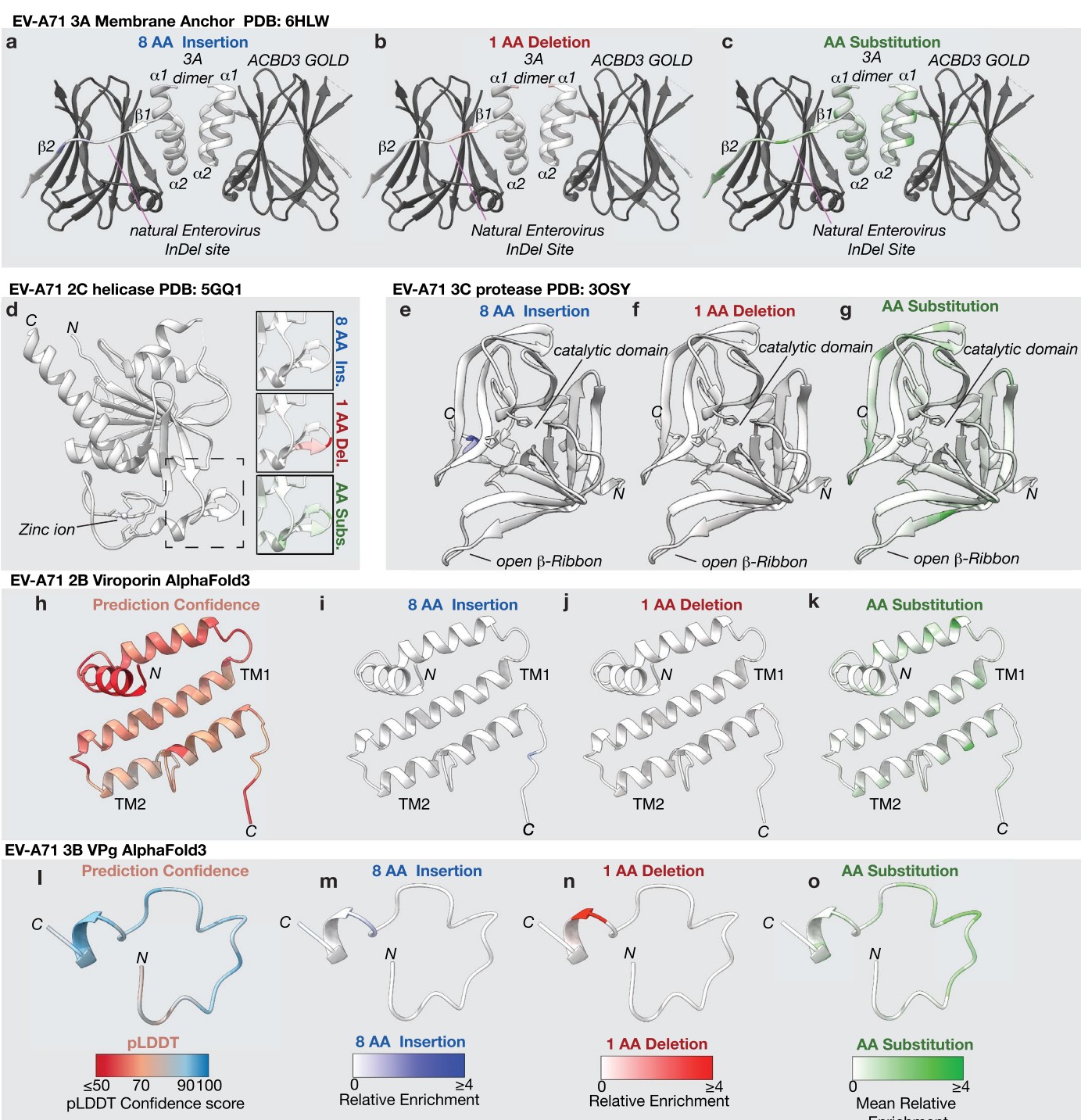

**Extended Data Fig. 6 | Structural interpretation of mutational effects for 3A, 2C, 3C(pro), 2B and 3B.** Relative enrichment scores of insertions (**a**), deletions (**b**), and AA changes (**c**) mapped onto the crystal structure of 3A in complex with the human ACBD3 GOLD domain (PDB: 6HLW). The crystal structure of 2C (PDB: 5GQ1) is shown in (**d**) with a zoom in on a loop displaying the relative enrichment scores for InDels and AA changes. The relative enrichment scores for insertions (**e**), deletions (**f**), and AA substitutions (**g**) mapped onto the structure of 3C(pro) (PDB: 3OSY). (**h**) AlphaFold3 prediction of EV-A71 2B with the pLDDT score mapped onto the predicted structure. The relative enrichment scores for (**i**) insertions, (**j**) deletions, and (**k**) AA substitutions were mapped onto the predicted structure of 2B. (**l**) AlphaFold3 prediction of EV-A71 3B with the pLDDT score mapped onto the predicted structure. The relative enrichment scores for insertions (**m**), deletions (**n**), and AA changes (**o**) were mapped onto the predicted structure of 3B.

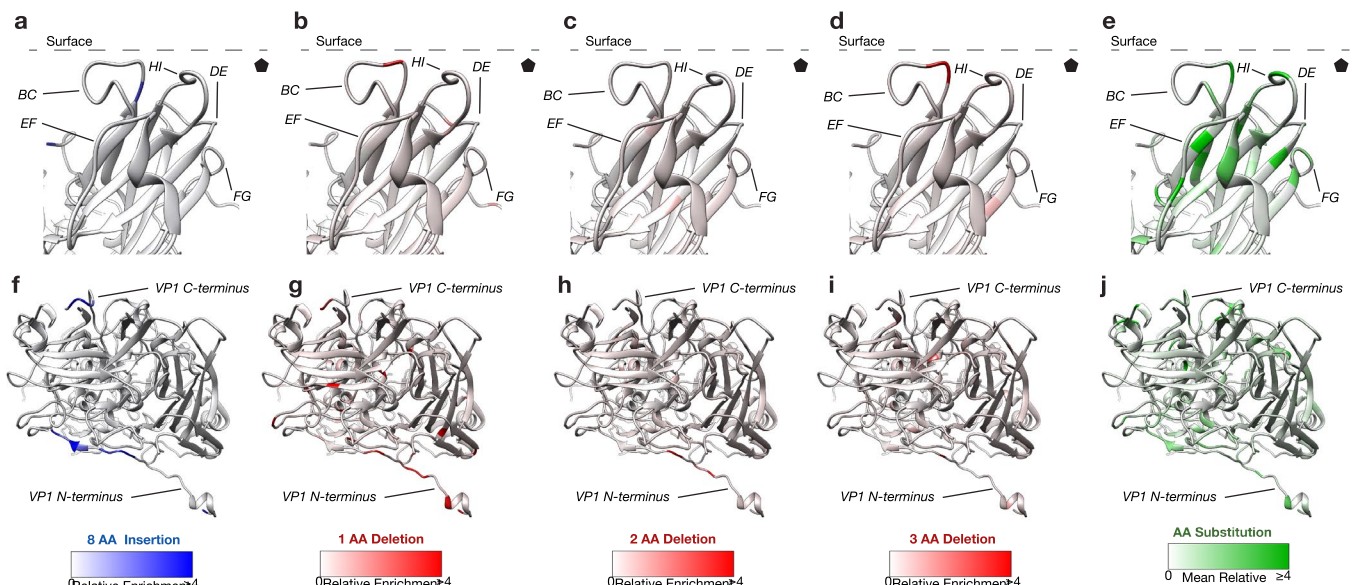

**Extended Data Fig. 7 | Structural interpretation of mutational effects of the Capsid.** Zoom on (**a**) 8AA Insertion, (**b**) 1 AA deletion, (**b**) 2 AA deletion, (**d**) 3 AA deletion, and (**e**) AA substitution hotspots from panels in Fig. 5 focusing on the surface-exposed loops of VP1. (**f**–**j**) Zoom in on same data to highlight variants enriched at the N- and C- termini of VP1: (**f**) 8AA Insertion, (**g**) 1 AA deletion, (**h**) 2 AA deletion, (**i**) 3 AA deletion, and (**j**) AA substitution hotspots. The PDB structure used for the EV-A71 virion was PDB: 8E2X.

# Reporting Summary

## Statistics

For all statistical analyses, confirm that the following items are present in the figure legend, table legend, main text, or Methods section.

| n/a | Confirmed | |
|---|---|---|
| ☐ | ☒ | The exact sample size (*n*) for each experimental group/condition, given as a discrete number and unit of measurement |
| ☐ | ☒ | A statement on whether measurements were taken from distinct samples or whether the same sample was measured repeatedly |
| ☐ | ☒ | The statistical test(s) used AND whether they are one- or two-sided<br>*Only common tests should be described solely by name; describe more complex techniques in the Methods section.* |
| ☒ | ☐ | A description of all covariates tested |
| ☐ | ☒ | A description of any assumptions or corrections, such as tests of normality and adjustment for multiple comparisons |
| ☐ | ☒ | A full description of the statistical parameters including central tendency (e.g. means) or other basic estimates (e.g. regression coefficient) AND variation (e.g. standard deviation) or associated estimates of uncertainty (e.g. confidence intervals) |
| ☐ | ☒ | For null hypothesis testing, the test statistic (e.g. $F$, $t$, $r$) with confidence intervals, effect sizes, degrees of freedom and $P$ value noted<br>*Give P values as exact values whenever suitable.* |
| ☒ | ☐ | For Bayesian analysis, information on the choice of priors and Markov chain Monte Carlo settings |
| ☒ | ☐ | For hierarchical and complex designs, identification of the appropriate level for tests and full reporting of outcomes |
| ☒ | ☐ | Estimates of effect sizes (e.g. Cohen's *d*, Pearson's *r*), indicating how they were calculated |

*Our web collection on statistics for biologists contains articles on many of the points above.*

## Software and code

Policy information about availability of computer code

| | |
|---|---|
| Data collection | To collect sequencing data and analyse, we used both commercial sequencing software (illumina bcl2fastq/2.20, guppy_gpu/6.0.6 or guppy/6.5.7. and minimap2/2.24 or minimap2/2.26) for mapping insertions or deletions, we used either stickleback (insertions), or deletionmapper0.2.py (deletions). These were both developed in house to analyse these data. These scripts are included in the repository included in the manuscript https://doi.org/10.5061/dryad.866t1g1xm. |
| Data analysis | Python scripts used in mapping the reads with engineered insertions and deletions are available at https://github.com/QVEU/InDel_Toolkit. The R scripts associated with specific bioinformatic computations performed herein are available at https://github.com/QVEU/eva71_dimple. Rscripts to regenerate all the figures are included in the dryad repository associated with this manuscript, along with all the python and R script versions used for this study: https://doi.org/10.5061/dryad.866t1g1xm.<br><br>R scripts for analysis and figure generation were run using the R version 4.0.3 (2020-10-10). R packages used were: ggplot2, tidyverse, tidyr, ggpubr, dplyr, ggridges, ineq, RColorBrewer, stringr, gglorenz, readr, scales, zoo, Biostrings, DescTools. AnalyzeSaturationMutagenesis in GATK version 4.2.6.0 or 4.5.0.0 was used for analysis of amino acid change experiments. The Enrich2 software (v1.3.1) was used for calculation of enrichment scores. Chimera sessions were created in Chimera production version 1.16 (build 42360). |

For manuscripts utilizing custom algorithms or software that are central to the research but not yet described in published literature, software must be made available to editors and reviewers. We strongly encourage code deposition in a community repository (e.g. GitHub). See the Nature Portfolio guidelines for submitting code & software for further information.

## Data

Policy information about availability of data

All manuscripts must include a data availability statement. This statement should provide the following information, where applicable:

- Accession codes, unique identifiers, or web links for publicly available datasets
- A description of any restrictions on data availability
- For clinical datasets or third party data, please ensure that the statement adheres to our policy

All raw sequencing read data is available in the SRA database under NCBI project number, PRJNA1066851. Structures used in this manuscript are available through these accession codes: PDB: 3W95 (EV-A71 2A), PDB: 5GQ1 (EV-A71 2C), PDB: 6HLW (EV-A71 3A), PDB: 3OSY (EV-A71 3C), PDB: 3N6L or 6KWQ (EV-A71 3D), PDB: 8E2X (EV-A71 virion), and PDB: 7QW9 (CV-A6 virion).

## Research involving human participants, their data, or biological material

Policy information about studies with human participants or human data. See also policy information about sex, gender (identity/presentation), and sexual orientation and race, ethnicity and racism.

| | |
|---|---|
| Reporting on sex and gender | NA |
| Reporting on race, ethnicity, or other socially relevant groupings | NA |
| Population characteristics | NA |
| Recruitment | NA |
| Ethics oversight | NA |

Note that full information on the approval of the study protocol must also be provided in the manuscript.

# Field-specific reporting

Please select the one below that is the best fit for your research. If you are not sure, read the appropriate sections before making your selection.

☒ Life sciences ☐ Behavioural & social sciences ☐ Ecological, evolutionary & environmental sciences

For a reference copy of the document with all sections, see nature.com/documents/nr-reporting-summary-flat.pdf

# Life sciences study design

All studies must disclose on these points even when the disclosure is negative.

| | |
|---|---|
| Sample size | Sample size was determined based on previous studies using deep-mutational scanning in Enteroviruses, and the depth and complexity of the libraries. Based on previous studies three biological replicates was sufficient to distinguish strong fitness effects. The studies used as a reference were: https://elifesciences.org/articles/64256 and https://journals.plos.org/plosbiology/article?id=10.1371/journal.pbio.3002709 |
| Data exclusions | No data were excluded from the analyses. |
| Replication | Data represent three biological replicates of each screen. Replication attempts were successful and showed high correlation. |
| Randomization | Randomization is not relevant in these studies. However, related to this idea, these studies involve large pools of mutants maintained at population sizes that ensure well-balanced representation (and verified by sequencing). Care is taken, through propagation to mitigate any bottlenecking which might bias results. These large screens are performed in biological replicate to identify sampling bias or error in replicate populations. |
| Blinding | No blinding was used or appropriate for this study. We deal with high-throughput sequencing data that includes analysis using standardized data analysis frameworks. Knowledge of samples does not affect outcome of the analysis. |

# Reporting for specific materials, systems and methods

We require information from authors about some types of materials, experimental systems and methods used in many studies. Here, indicate whether each material, system or method listed is relevant to your study. If you are not sure if a list item applies to your research, read the appropriate section before selecting a response.

## Materials & experimental systems

| n/a | Involved in the study |
|---|---|
| ☒ ☐ | Antibodies |
| ☐ ☒ | Eukaryotic cell lines |
| ☒ ☐ | Palaeontology and archaeology |
| ☒ ☐ | Animals and other organisms |
| ☒ ☐ | Clinical data |
| ☒ ☐ | Dual use research of concern |
| ☒ ☐ | Plants |

## Methods

| n/a | Involved in the study |
|---|---|
| ☒ ☐ | ChIP-seq |
| ☒ ☐ | Flow cytometry |
| ☒ ☐ | MRI-based neuroimaging |

## Eukaryotic cell lines

Policy information about cell lines and Sex and Gender in Research

| | |
|---|---|
| Cell line source(s) | Source: American Type Culture Collection (ATCC). Cell line used is Rhabdomyosarcoma (RD) (ATCC, CCL-136) |
| Authentication | Cells have been characterized by genotyping, confirming them as Rhabdomyosarcoma cells. |
| Mycoplasma contamination | All cells tested negative for mycoplasma contamination by isothermal amplification prior to the experiment. |
| Commonly misidentified lines (See ICLAC register) | Cell lines are not listed in the database. Cell lines verified genetically as RD, rhabdomyosarcoma. Tested cells were from lot and passage of those used in this study. |

## Plants

| | |
|---|---|
| Seed stocks | *Report on the source of all seed stocks or other plant material used. If applicable, state the seed stock centre and catalogue number. If plant specimens were collected from the field, describe the collection location, date and sampling procedures.* |
| Novel plant genotypes | *Describe the methods by which all novel plant genotypes were produced. This includes those generated by transgenic approaches, gene editing, chemical/radiation-based mutagenesis and hybridization. For transgenic lines, describe the transformation method, the number of independent lines analyzed and the generation upon which experiments were performed. For gene-edited lines, describe the editor used, the endogenous sequence targeted for editing, the targeting guide RNA sequence (if applicable) and how the editor was applied.* |
| Authentication | *Describe any authentication procedures for each seed stock used or novel genotype generated. Describe any experiments used to assess the effect of a mutation and, where applicable, how potential secondary effects (e.g. second site T-DNA insertions, mosiacism, off-target gene editing) were examined.* |

