## [Peer Review File · Nature Microbiology]

Deep mutation, insertion, and deletion scanning across the Enterovirus A proteome reveals constraints shaping viral evolution

Corresponding Author: Dr Patrick Dolan

Version 0:

Reviewer comments:

Reviewer #1

(Remarks to the Author)

Summary:

The manuscript by Bakhache et al. describes the results of a large InDel screen on Enterovirus A71. The authors find that defined regions in the viral proteome are more prone to tolerate insertions or deletions of amino acids with hotspots in the viral protease 2A(pro) and in parts of the capsid protein VP1. These regions match with evolutionary less conserved sequence spaces. The authors go on to map the regions that tolerate InDels to available structural data and also try to predict the effect of two InDels on protein structure using AlphaFold.

The manuscript is very well-written, concise, and easily understandable. In the current version, however, the authors do only limited analysis of the potential structural effects of these InDels which is not sufficient.

Major concerns:

-The analysis depicted in Fig. 4, is currently very limited as only one protein (2A(pro)) is analyzed. This analysis of which structural features tolerate InDels should be done for all A71 proteins to make the conclusions significant. If experimental structures do not exist, AlphaFold predictions can be used if their scores (MSA depth, pLDDT, PAE) indicate well-predicted structures. (Please refer to the original publication by Jumper et al. and the DeepMind/EMBL AlphaFold database FAQ for guidance if needed.)

Using AlphaFold predictions for all A71 proteins would also have the benefit that the pLDDT values would indicate flexible disordered regions (pLDDT below 50), and the PAE data would indicate structural flexibility between domains. These areas should be enriched for InDel tolerance according to the conclusions of this manuscript. Please also see my last comment below on Fig 5M-O in this regard.

-Based on the current analysis, it is unclear why most reported InDels are not tolerated. Instead of providing few examples, the authors should provide a more exhaustive analysis of the likely effect on protein stability (e.g. $\Delta\Delta G$ stability changes using Rosetta, see <https://www.nature.com/articles/s41594-022-00849-w>), active sites and protein-protein interactions (e.g. using AlphaFold-Multimer v2.3 and scoring by ipTM).

-Fig 5M-O: From the provided data (the linked Dryad repository wasn't working for me), it is unclear how good the VP1 prediction is. The authors should provide MSA, pLDDT, PAE and pTM data for reference. Moreover, whether VP1 was predicted as a monomer or pentamer is unclear. To be able to compare to the pentamer structure in 5A-L, the predictions would need to be run as a pentamer (e.g., using local ColabFold in Multimer mode and enough GPU RAM and enough cycles until the model pLDDTs converge). This might also change the position of the EF loop, which is placed differently in the current prediction (compare Fig 5I-L with 5 M-O).

-Fig 5M-O: to evaluate the utility of AlphaFold predictions to detect the effect of InDels, the authors need to predict at least a representative number of structures with deleterious and tolerated InDels and quantify their effect not only on local structure but also on global protein folding either using approaches such as those described here: <https://journals.aps.org/prl/abstract/10.1103/PhysRevLett.131.218401> or the link provided above. Finally, when evaluating loop positions, as shown in Fig 5M-O, the PAE and pLDDT values need to be plotted to ensure that the predicted loop position is not ambiguous due to low scores.

Minor:

The last sentence of the second paragraph in the introduction is not grammatically correct.

Reviewer #2

(Remarks to the Author)

This manuscript by Bakhache et al employs deep InDel scanning on the Enterovirus A71 capsid protein to explore the structural tolerance to insertion and deletions mutations. InDels allow massive jumps in sequence space, but they are unexplored experimentally due to technical challenges associated with making InDels. The authors used recently developed pipelines that solve these problems, SPINE and DIMPLE, that allow systematic efficient assembly of InDel libraries. Taken together, the study is amongst the first systematic surveys of insertions and deletions on a protein and currently (to my knowledge) is the largest protein such approaches have been taken. The authors do an excellent job of comparing the results of the variants to the structural properties of the viral capsid protein. The authors did not have sufficient technical details to follow how the screens were conducted and how data was processed. Additionally, it would be nice to see further discussion of the results of the study with prior studies. Due to the novelties of the studies, yet necessary changes to improve the manuscript, I recommend a revision at Nature Microbiology.

Major Concerns

1. While this study is highly innovative in that it conducts an indel scan on a massive protein. While it would have been nice to see missense variants alongside indels to aid in interpretation, the results here could be contextualized with those in prior studies solely focused on indels. The manuscript will likely be of interest to others who work on indels and as such it could be good to discuss the tolerance seen in this manuscript contextualized with

other studies and what secondary structural elements are differentially sensitive to insertions and deletions. Some manuscripts the authors may want to cite and discuss include a landmark AAV engineering manuscript generated by oligo pools (Ogden, 2019, Science), antibiotic resistance protein (Gonzalez CE, 2019), a recent domain insert on AAV, an indel scan within the beta amyloid protein (Seuma, Nature Comm, 2023), a systematic insertional studies in a potassium channel that included many more inserted motifs using this platform (Coyote-Maestas, Nature Comm, 2022), and a recent preprint that conducted mutational scans across many unrelated proteins (Topolska, biorxiv, 2023).

2. The authors use AlphaFold2 to predict a structural model of VP1 (8E2X) capsid proteins including deletions (Fig 5M-Q). While I find it highly interesting to explore the structural consequences of deletions I would urge major caution in interpreting or including this model in the manuscript. From my reading this model is not essential to the manuscript and indel effects on protein structures are under-explored by structural biologists. AlphaFold2 was trained on the protein data bank and multiple sequence alignments, which both don't typically handle indels very much. Evidence of indels not being well-handled by AlphaFold2 are that the unstructured regions of proteins have low accuracy of prediction in AlphaFold2 and the associated variant effect predictor Alpha missense cannot predict indels (Cheng J, 2023, Science). More broadly, AlphaFold2 struggles to predict the effects of mutations alone (Pak MA, et al PLoS One, 2023). If the authors do want a structural model, something that incorporates physics would be worthwhile such as molecular dynamics simulations after a perturbation is made. For example, one can include MD simulations after an AlphaFold2-like structure prediction effort to relax the ensemble (as in Yee SW, Macdonald CB, 2023, Biorxiv), however that has not been validated well for deletions. My suggestion is to not include these models as they are not essential to the manuscript and do not add much beyond simply mapping the indel scores while implying incorrectly that structural models of indels exist.

3. The methods section of the manuscript is missing many details that are necessary to follow what was done in the screens and what the computational pipelines that were used for analyzing the data. The screen is mentioned in the 'Virus Passaging and Sequencing' section however it is unclear how many passages were done and how long those passages were. Perhaps adding more detail in the workflow cartoon in Figure 2B would be helpful for understanding the experiments as well. Similarly, the fitness scores are currently not mentioned within the method section with the description of data processing ends after alignment prior to going to structural analysis. It is difficult to judge the technical aspects of the indel scanning work and others would likely need more details if they wanted to conduct similar experiments.

Minor Concerns

1. Please add line numbers they help with reviewing and making specific suggestions.

2. The authors mention that the libraries vary by a modest three and six fold range in insertions and deletions. Was there any systematic pattern to variance? Were shorter vs longer sequences differentially selected? Including this information would be useful for other people who are conducting indel scanning and for improvement for library generation pipelines to reduce bias.

3. From my reading the authors conducted two separate screens for insertions and deletions. They do well to only make comparisons qualitatively across these groups. However, in Figure 2A these are put on the same plot which is misleading. I recommend separating these.

4. Figure 2A- The insertions side of the plot is missing a dotted line as 'neutral insertions' that the neutral deletion has'.

5. The authors do an interesting experiment where they insert highly variable 5-amino acid peptides. Why was a 5 amino acid peptide used? A bit more rationale here could be useful. From my reading the peptide was of a single amino acid type, however the wording in 'The inserted sequence consisted of each of the 20 amino acids flanked by flexible linkers' is a bit confusing. Specifically what flexible linkers?

6. In figure 3 D-G the authors compare the physicochemical properties of the peptides within specific regions of the proteins using stacked barplots. The authors descriptions in the text are hard to follow as they do not refer to specific regions based on amino acid sequences. From positions ~565-575 Y seem to dominate whereas positions 585-595 K and N seem important. Some more context and discussion here could be useful. It may also be useful to plot just these subregions as heatmaps which would allow easier interpretation. Could also be useful highlighting these regions in Figure 4.

7. Figure 5F-H shows relative fitness mapped onto the external surface of the capsid. In this it's abundantly clear from figs 5F and 5H that there are hotspots that have positive fitness for deletions 1 and 3 amino acid long. However, the subpanel of 2AA deletions Fig 5G contains no visible hotspots. Are deletions 1 or 3 Amino acids better tolerated than 2 Amino acid deletions? I find that surprising - why might this be?

8. Figure 5I-L shows many subpanels to represent the effects of indels across the structural of the virus. I find it difficult to see exactly what the authors are discussing the text. While there are really interesting observed trends such as, 'Another interesting observation centered around two antiparallel Beta-strands, with B-strand e and f tolerating a 2 AA and 3 AA deletion.' While if I squint I can see the deletions 'E and F' there are many other motifs shown. For example two aromatics are shown in stick mode as is a substrate or small molecule of sorts which likely is not necessary for the interpretation. Adjusting the camera shading in chimera or pymol can reduce the noise.

Willow Coyote-Maestas

Reviewer #3

(Remarks to the Author)

Bakhache and colleagues present an innovative method to probe the effects of small insertions and deletions across the single polypeptide of EV-A71. Extending DMS methods beyond amino acid substitutions is an exciting advance.

However, I am left a little unsure what exactly I have learned. This is partly due to the fact that the space of InDel variation is vastly bigger than that of substitutions and it is thus harder to interpret the observations. But I also struggle with getting a sense of how accurate the measurements are, what it means for an insertion to be very beneficial etc. I list a few specific points that I would like to see clarified or addressed. In particular, I think additional validation of the experimental assay would make the paper a lot stronger.

Insertion tolerance: I am reading Fig 2A correctly that non of the insertions to VP2 that seem tolerated or even beneficial in passage 1 survive passage 2? Does this suggest that results from passage 1 are affected by complementation or interference in a major way? Also, why only show the first third of the genome?

It would be more helpful to plot a quantity like 'gap density' along the EV A71 reference genome and integrate it into Fig 2A. Fig 2B is currently hard to compare with 2A because it is slightly stretched relative to the reference genome.

Distribution of fitness effects: One of the main results of this work is a quantification of what fraction of InDels are tolerated, are deleterious, or beneficial. Somewhat unsurprisingly, the great majority of such perturbations are lethal. What is more surprising, is how often a InDel is classified as beneficial. In fact, almost no InDels are neutral, while in some regions 10 or even 20% are beneficial. This makes me wonder how reliable this classification of non-lethal mutations into beneficial, neutral, and deleterious is. Did the authors quantify the accuracy of these fitness assignments in competition of individual mutants with the wildtype (or maybe single genotype growth curves to rule out complementation)? Such validation would increase the confidence in these results substantially.

In the section on phylogenetic analysis, I am not sure the statement "...allowed us to understand the emergence of InDels across EV-A species evolution" is warranted. You have described length variation among EV-A genomes and these results are consistent with the DMS results. But these results are descriptive rather than explanatory.

Is it known whether they InDels in the termini of VP1 affect cleavage between VP1 and VP3 and 2A? I remember discussion about the ambiguous cleave site locations between VP3 and VP1 in EV-D 68 and if similar variation exists for EV-A71, it might affect the structural interpretation of the insertions discussed here.

Minor suggestions:

Introduction: I felt the introduction a bit too black and white in the way it is contrasting nonsynonymous mutations and indels. Almost every major SC2 variant was characterized by specific indels in the N terminal domain of spike. Likewise, HIV-1 has frequent indel variation in its variable loops, sometimes even compensated frameshifts.

Fig 2: the bar graph is hard to read. the majority of points are not visible since they correspond to lethal mutations. The entire deleterious range is compressed into the narrow 0 to 1 interval. I am wondering whether plotting these data on a logarithmic scale with a separate bin for lethal would do the data more justice. The same applied to Fig 2C&D: almost none of the relevant data can be seen. Here, I'd recommend a cumulative histogram with logarithmic axes.

Richard Neher

Decision Letter:

28th February 2024

Dear Patrick,

Thank you for your patience while your manuscript "Uncovering Structural Plasticity of Enterovirus A through Deep Insertional and Deletional Scanning" was under peer-review at Nature Microbiology. It has now been seen by 3 referees, whose expertise and comments you will find at the end of this email. Although they find your work of some potential interest, they have raised a number of concerns that will need to be addressed before we can consider publication of the work in Nature Microbiology.

In particular, reviewer #1 requests to extend your analyses and reviewer #2 asks to better explain the technical aspects of your study. Please also clarify the advance of your findings and validate your results using experimental data, as requested by reviewer #3.

Should further experimental data allow you to address these criticisms, we would be happy to look at a revised manuscript.

Please include a data availability statement as a separate section after Methods but before references, under the heading "Data Availability". This section should inform readers about the availability of the data used to support the conclusions of your study. This information includes accession codes to public repositories (data banks for protein, DNA or RNA sequences, microarray, proteomics data etc...), references to source data published alongside the paper, unique identifiers such as URLs to data repository entries, or data set DOIs, and any other statement about data availability. At a minimum, you should include the following statement: "The data that support the findings of this study are available from the corresponding author upon request", mentioning any restrictions on availability. If DOIs are provided, we also strongly encourage including these in the Reference list (authors, title, publisher (repository name), identifier, year). For more guidance on how to write this section please see: <http://www.nature.com/authors/policies/data/data-availability-statements-data-citations.pdf>

* If you have not done so already we suggest that you begin to revise your manuscript so that it conforms to our Article format instructions at <http://www.nature.com/nmicrobiol/info/final-submission>. Refer also to any guidelines provided in this letter.

When submitting the revised version of your manuscript, please pay close attention to our [href="https://www.nature.com/nature-portfolio/editorial-policies/image-integrity">Digital Image Integrity Guidelines. and to the following points below:](https://www.nature.com/nature-portfolio/editorial-policies/image-integrity)

Link Redacted

Note: This url links to your confidential homepage and associated information about manuscripts you may have submitted or be reviewing for us. If you wish to forward this e-mail to co-authors, please delete this link to your homepage first.

Nature Microbiology is committed to improving transparency in authorship. As part of our efforts in this direction, we are now requesting that all authors identified as 'corresponding author' on published papers create and link their Open Researcher and Contributor Identifier (ORCID) with their account on the Manuscript Tracking System (MTS), prior to acceptance. This applies to primary research papers only. ORCID helps the scientific community achieve unambiguous attribution of all scholarly contributions. You can create and link your ORCID from the home page of the MTS by clicking on 'Modify my Springer Nature account'. For more information please visit www.springernature.com/orcid.

If you wish to submit a suitably revised manuscript we would hope to receive it within 6 months. If you cannot send it within this time, please let us know. We will be happy to consider your revision, even if a similar study has been accepted for publication at Nature Microbiology or published elsewhere (up to a maximum of 6 months).

Yours sincerely,

Reviewer Expertise:

Referee #1: Virology, Proteome, Structure

Referee #2: Deep mutational scanning

Referee #3: Virus evolution

Reviewer Comments:

Reviewer #1 (Remarks to the Author):

Summary:

The manuscript by Bakhache et al. describes the results of a large InDel screen on Enterovirus A71. The authors find that defined regions in the viral proteome are more prone to tolerate insertions or deletions of amino acids with hotspots in the viral protease 2A(pro) and in parts of the capsid protein VP1. These regions match with evolutionary less conserved sequence spaces. The authors go on to map the regions that tolerate InDels to available structural data and also try to predict the effect of two InDels on protein structure using AlphaFold.

The manuscript is very well-written, concise, and easily understandable. In the current version, however, the authors do only limited analysis of the potential structural effects of these InDels which is not sufficient.

Major concerns:

-The analysis depicted in Fig. 4, is currently very limited as only one protein (2A(pro)) is analyzed. This analysis of which structural features tolerate InDels should be done for all A71 proteins to make the conclusions significant. If experimental structures do not exist, AlphaFold predictions can be used if their scores (MSA depth, pLDDT, PAE) indicate well-predicted structures. (Please refer to the original publication by Jumper et al. and the DeepMind/EMBL AlphaFold database FAQ for guidance if needed.)

Using AlphaFold predictions for all A71 proteins would also have the benefit that the pLDDT values would indicate flexible disordered regions (pLDDT below 50), and the PAE data would indicate structural flexibility between domains. These areas should be enriched for InDel tolerance according to the conclusions of this manuscript. Please also see my last comment below on Fig 5M-O in this regard.

-Based on the current analysis, it is unclear why most reported InDels are not tolerated. Instead of providing few examples, the authors should provide a more exhaustive analysis of the likely effect on protein stability (e.g. $\Delta\Delta G$ stability changes using Rosetta, see <https://www.nature.com/articles/s41594-022-00849-w>), active sites and protein-protein interactions (e.g. using AlphaFold-Multimer v2.3 and scoring by ipTM).

-Fig 5M-O: From the provided data (the linked Dryad repository wasn't working for me), it is unclear how good the VP1 prediction is. The authors should provide MSA, pLDDT, PAE and pTM data for reference. Moreover, whether VP1 was predicted as a monomer or pentamer is unclear. To be able to compare to the pentamer structure in 5A-L, the predictions would need to be run as a pentamer (e.g., using local ColabFold in Multimer mode and enough GPU RAM and enough cycles until the model pLDDTs converge). This might also change the position of the EF loop, which is placed differently in the current prediction (compare Fig 5I-L with 5 M-O).

-Fig 5M-O: to evaluate the utility of AlphaFold predictions to detect the effect of InDels, the authors need to predict at least a representative number of structures with deleterious and tolerated InDels and quantify their effect not only on local structure but also on global protein folding either using approaches such as those described here: <https://journals.aps.org/prl/abstract/10.1103/PhysRevLett.131.218401> or the link provided above. Finally, when evaluating loop positions, as shown in Fig 5M-O, the PAE and pLDDT values need to be plotted to ensure that the predicted loop position is not ambiguous due to low scores.

Minor:

The last sentence of the second paragraph in the introduction is not grammatically correct.

Reviewer #2 (Remarks to the Author):

This manuscript by Bakhache et al employs deep InDel scanning on the Enterovirus A71 capsid protein to explore the structural tolerance to insertion

and deletions mutations. InDels allow massive jumps in sequence space, but they are unexplored experimentally due to technical challenges associated with making InDels. The authors used recently developed pipelines that solve these problems, SPINE and DIMPLE, that allow systematic efficient assembly of InDel libraries. Taken together, the study is amongst the first systematic surveys of insertions and deletions on a protein and currently (to my knowledge) is the largest protein such approaches have been taken. The authors do an excellent job of comparing the results of the variants to the structural properties of the viral capsid protein. The authors did not have sufficient technical details to follow how the screens were conducted and how data was processed. Additionally, it would be nice to see further discussion of the results of the study with prior studies. Due to the novelties of the studies, yet necessary changes to improve the manuscript, I recommend a revision at Nature Microbiology.

Major Concerns

1. While this study is highly innovative in that it conducts an indel scan on a massive protein. While it would have been nice to see missense variants alongside indels to aid in interpretation, the results here could be contextualized with those in prior studies solely focused on indels. The manuscript will likely be of interest to others who work on indels and as such it could be good to discuss the tolerance seen in this manuscript contextualized with other studies and what secondary structural elements are differentially sensitive to insertions and deletions. Some manuscripts the authors may want to cite and discuss include a landmark AAV engineering manuscript generated by oligo pools (Ogden, 2019, Science), antibiotic resistance protein (Gonzalez CE, 2019), a recent domain insert on AAV, an indel scan within the beta amyloid protein (Seuma, Nature Comm, 2023), a systematic insertional studies in a potassium channel that included many more inserted motifs using this platform (Coyote-Maestas, Nature Comm, 2022), and a recent preprint that conducted mutational scans across many unrelated proteins (Topolska, biorxiv, 2023).

2. The authors use AlphaFold2 to predict a structural model of VP1 (8E2X) capsid proteins including deletions (Fig 5M-Q). While I find it highly interesting to explore the structural consequences of deletions I would urge major caution in interpreting or including this model in the manuscript. From my reading this model is not essential to the manuscript and indel effects on protein structures are under-explored by structural biologists. AlphaFold2 was trained on the protein data bank and multiple sequence alignments, which both don't typically handle indels very much. Evidence of indels not being well-handled by AlphaFold2 are that the unstructured regions of proteins have low accuracy of prediction in AlphaFold2 and the associated variant effect predictor Alpha missense cannot predict indels (Cheng J, 2023, Science). More broadly, AlphaFold2 struggles to predict the effects of mutations alone (Pak MA, et al PLoS One, 2023). If the authors do want a structural model, something that incorporates physics would be worthwhile such as molecular dynamics simulations after a perturbation is made. For example, one can include MD simulations after an AlphaFold2-like structure prediction effort to relax the ensemble (as in Yee SW, Macdonald CB, 2023, Biorxiv), however that has not been validated well for deletions. My suggestion is to not include these models as they are not essential to the manuscript and do not add much beyond simply mapping the indel scores while implying incorrectly that structural models of indels exist.

3. The methods section of the manuscript is missing many details that are necessary to follow what was done in the screens and what the computational pipelines that were used for analyzing the data. The screen is mentioned in the 'Virus Passaging and Sequencing' section however it is unclear how many passages were done and how long those passages were. Perhaps adding more detail in the workflow cartoon in Figure 2B would be helpful for understanding the experiments as well. Similarly, the fitness scores are currently not mentioned within the method section with the description of data processing ends after alignment prior to going to structural analysis. It is difficult to judge the technical aspects of the indel scanning work and others would likely need more details if they wanted to conduct similar experiments.

Minor Concerns

1. Please add line numbers they help with reviewing and making specific suggestions.

2. The authors mention that the libraries vary by a modest three and six fold range in insertions and deletions. Was there any systematic pattern to variance? Were shorter vs longer sequences differentially selected? Including this information would be useful for other people who are conducting indel scanning and for improvement for library generation pipelines to reduce bias.

3. From my reading the authors conducted two separate screens for insertions and deletions. They do well to only make comparisons qualitatively across these groups. However, in Figure 2A these are put on the same plot which is misleading. I recommend separating these.

4. Figure 2A- The insertions side of the plot is missing a dotted line as 'neutral insertions' that the neutral deletion has'.

5. The authors do an interesting experiment where they insert highly variable 5-amino acid peptides. Why was a 5 amino acid peptide used? A bit more rationale here could be useful. From my reading the peptide was of a single amino acid type, however the wording in 'The inserted sequence consisted of each of the 20 amino acids flanked by flexible linkers' is a bit confusing. Specifically what flexible linkers?

6. In figure 3 D-G the authors compare the physicochemical properties of the peptides within specific regions of the proteins using stacked barplots. The authors descriptions in the text are hard to follow as they do not refer to specific regions based on amino acid sequences. From positions ~565-575 Y seem to dominate whereas positions 585-595 K and N seem important. Some more context and discussion here could be useful. It may also be useful to plot just these subregions as heatmaps which would allow easier interpretation. Could also be useful highlighting these regions in Figure 4.

7. Figure 5F-H shows relative fitness mapped onto the external surface of the capsid. In this it's abundantly clear from figs 5F and 5H that there are hotspots that have positive fitness for deletions 1 and 3 amino acid long. However, the subpanel of 2AA deletions Fig 5G contains no visible hotspots. Are deletions 1 or 3 Amino acids better tolerated than 2 Amino acid deletions? I find that surprising - why might this be?

8. Figure 5I-L shows many subpanels to represent the effects of indels across the structural of the virus. I find it difficult to see exactly what the authors are discussing the text. While there are really interesting observed trends such as, 'Another interesting observation centered around two antiparallel Beta-strands, with B-strand e and f tolerating a 2 AA and 3 AA deletion.' While if I squint I can see the deletions 'E and F' there are many other motifs shown. For example two aromatics are shown in stick mode as is a substrate or small molecule of sorts which likely is not necessary for the interpretation. Adjusting the camera shading in chimera or pymol can reduce the noise.

Willow Coyote-Maestas

Reviewer #3 (Remarks to the Author):

Bakhache and colleagues present an innovative method to probe the effects of small insertions and deletions across the single polypeptide of EV-A71. Extending DMS methods beyond amino acid substitutions is an exciting advance.

However, I am left a little unsure what exactly I have learned. This is partly due to the fact that the space of InDel variation is vastly bigger than that of substitutions and it is thus harder to interpret the observations. But I also struggle with getting a sense of how accurate the measurements are, what it means for an insertion to be very beneficial etc. I list a few specific points that I would like to see clarified or addressed. In particular, I think additional validation of the experimental assay would make the paper a lot stronger.

Insertion tolerance: I am reading Fig 2A correctly that non of the insertions to VP2 that seem tolerated or even beneficial in passage 1 survive

passage 2? Does this suggest that results from passage 1 are affected by complementation or interference in a major way? Also, why only show the first third of the genome?

It would be more helpful to plot a quantity like 'gap density' along the EV A71 reference genome and integrate it into Fig 2A. Fig 2B is currently hard to compare with 2A because it is slightly stretched relative to the reference genome.

Distribution of fitness effects: One of the main results of this work is a quantification of what fraction of InDels are tolerated, are deleterious, or beneficial. Somewhat unsurprisingly, the great majority of such perturbations are lethal. What is more surprising, is how often a InDel is classified as beneficial. In fact, almost no InDels are neutral, while in some regions 10 or even 20% are beneficial. This makes me wonder how reliable this classification of non-lethal mutations into beneficial, neutral, and deleterious is. Did the authors quantify the accuracy of these fitness assignments in competition of individual mutants with the wildtype (or maybe single genotype growth curves to rule out complementation)? Such validation would increase the confidence in these results substantially.

In the section on phylogenetic analysis, I am not sure the statement "...allowed us to understand the emergence of InDels across EV-A species evolution" is warranted. You have described length variation among EV-A genomes and these results are consistent with the DMS results. But these results are descriptive rather than explanatory.

Is it known whether they InDels in the termini of VP1 affect cleavage between VP1 and VP3 and 2A? I remember discussion about the ambiguous cleave site locations between VP3 and VP1 in EV-D 68 and if similar variation exists for EV-A71, it might affect the structural interpretation of the insertions discussed here.

Minor suggestions:

Introduction: I felt the introduction a bit too black and white in the way it is contrasting nonsynonymous mutations and indels. Almost every major SC2 variant was characterized by specific indels in the N terminal domain of spike. Likewise, HIV-1 has frequent indel variation in its variable loops, sometimes even compensated frameshifts.

Fig 2: the bar graph is hard to read. the majority of points are not visible since they correspond to lethal mutations. The entire deleterious range is compressed into the narrow 0 to 1 interval. I am wondering whether plotting these data on a logarithmic scale with a separate bin for lethal would do the data more justice. The same applied to Fig 2C&D: almost none of the relevant data can be seen. Here, I'd recommend a cumulative histogram with logarithmic axes.

Richard Neher

Version 1:

Reviewer comments:

Reviewer #1

(Remarks to the Author)

Bakhache et al. provide a well-written and detailed revision of their manuscript. One substantial change in this version is the removal of all structure prediction data due to different recommendation of reviewer 2 and myself on the general topic. To evaluate this decision at least two different use cases need to be distinguished for which structural predictions would be useful:

1) As a "scaffold" to map regions of higher InDel tolerance onto proteins with no deposited experimental structure and determine in which secondary structural elements InDels and substitutions are better tolerated as done in Fig. 4, 5, and S8. From my viewpoint, there are no objections in the field right now to using structural predictions in the absence of experimental data for this kind of question as long as the models are well-validated along their scores (pLDDT, PAE, and pTM). An already 1,5 years old community assessment states: "In line with the reported high accuracy of the models, we found that AF2-predicted structures, on average, tend to give results that are as good as those derived from experimental structures." (<https://www.nature.com/articles/s41594-022-00849-w>). I also respectfully disagree with reviewer 2's conclusion about AlphaFold 2 showing low accuracy in unstructured regions, in case understood his point correctly. Intrinsically disordered regions should have low localization accuracy by definition. Moreover, the presence of IDRs is generally predicted extremely well by AF2. So well that protein disorder databases such as MobiDB rely on AF2 to detect IDRs by their low pLDDT. Also, if a secondary structure has a high pLDDT in a generally well-folded prediction, I do not see any issue in calling this feature to evaluate InDel frequency. Based on this, I do not see any reason not to use structural predictions for the remaining viral proteins as long as the scores are provided. In my opinion, including them, at least as a reference for future work, would be very useful.

2) To predict the effect of changes in regards to protein stability. Reviewer 2 and the authors are correct that this is a point of some debate in the field. However there is mounting evidence despite some negative reports that AF2 and now AF3 in combination with other methods such as "physics-based" modelling do work in evaluating the effect of sequence changes on protein stability and folding. However, I agree with the authors point that this might be out of the scope of this manuscript.

In conclusion, the authors have put a lot of extra work and effort into analysing the available experimental structures which makes the manuscript much stronger. I would appreciate if the authors would include AF2 or AF3 predictions of the remaining proteins for which no experimental structures are available, also to complete the analysis and provide a reference for future use.

Otherwise I do not have any further objections and would like to congratulate the authors on this very well-designed study and the very well-written paper.

Reviewer #2

(Remarks to the Author)

Fantastic job revising the manuscript. I appreciate all the changes the authors make and believe it to be an impactful manuscript with broad interest!

Willow Coyote-Maestas

Reviewer #3

(Remarks to the Author)

The authors have submitted an extensive revision of their manuscript and that address most points raised in my previous review. I think high throughput methodologies to assess the phenotypic impact of indel variation are important and in this light, this manuscript makes an important contribution. It is nice to see the correspondence between those locations with indel diversity among difference EV-A viruses.

remaining comment:

- the correlation of replicate results for indels in supp fig 5 D&E have some strange off-diagonal lines. Furthermore, why is the correlation more or less exact for some parts of the graphs for InDels, while one gets a more expected scatter for the substitutions?

Decision Letter:

16th July 2024

Dear Dr Dolan,

Thank you for your patience while your manuscript "Comprehensive Scanning of Insertions, Deletions, and Substitutions Reveals the Constraints Shaping Long-Term Diversification of Enterovirus A" was under peer-review at Nature Microbiology. It has now been seen by 3 referees, whose expertise and comments you will find at the of this email. You will see from their comments below that while they find your work of interest, some important points are raised. We are very interested in the possibility of publishing your study in Nature Microbiology, but would like to consider your response to these concerns in the form of a revised manuscript before we make a final decision on publication.

In particular, you will see that reviewer #1 requests to include the protein prediction data for proteins where no structures are available. While we understand that you removed the data due to the conflicting reports by reviewer #1 and #2, we find the arguments made by reviewer #1 convincing. The rest referees' reports are clear and the remaining issues should be straightforward to address.

If you have not done so already please begin to revise your manuscript so that it conforms to our Article format instructions at <http://www.nature.com/nmicrobiol/info/final-submission/>

The usual length limit for a Nature Microbiology Article is six display items (figures or tables) and 3,000 words. We have some flexibility, and can allow a revised manuscript at 3,500 words, but please consider this a firm upper limit. There is a trade-off of ~250 words per display item, so if you need more space, you could move a Figure or Table to Supplementary Information.

Some reduction could be achieved by focusing any introductory material and moving it to the start of your opening 'bold' paragraph, whose function is to outline the background to your work, describe in a sentence your new observations, and explain your main conclusions. The discussion should also be limited. Methods should be described in a separate section following the discussion, we do not place a word limit on Methods.

Nature Microbiology titles should give a sense of the main new findings of a manuscript, and should not contain punctuation. Please keep in mind that we strongly discourage active verbs in titles, and that they should ideally fit within 90 characters each (including spaces).

Please include a data availability statement as a separate section after Methods but before references, under the heading "Data Availability". This section should inform readers about the availability of the data used to support the conclusions of your study. This information includes accession codes to public repositories (data banks for protein, DNA or RNA sequences, microarray, proteomics data etc...), references to source data published alongside the paper, unique identifiers such as URLs to data repository entries, or data set DOIs, and any other statement about data availability. At a minimum, you should include the following statement: "The data that support the findings of this study are available from the corresponding author upon request", mentioning any restrictions on availability. If DOIs are provided, we also strongly encourage including these in the Reference list (authors, title, publisher (repository name), identifier, year). For more guidance on how to write this section please see: <http://www.nature.com/authors/policies/data/data-availability-statements-data-citations.pdf>

To improve the accessibility of your paper to readers from other research areas, please pay particular attention to the wording of the paper's opening bold paragraph, which serves both as an introduction and as a brief, non-technical summary in about 150 words. If, however, you require one or two extra sentences to explain your work clearly, please include them even if the paragraph is over-length as a result. The opening paragraph should not contain references. Because scientists from other sub-disciplines will be interested in your results and their implications, it is important to explain essential but specialised terms concisely. We suggest you show your summary paragraph to colleagues in other fields to uncover any problematic concepts.

If your paper is accepted for publication, we will edit your display items electronically so they conform to our house style and will reproduce clearly in print. If necessary, we will re-size figures to fit single or double column width. If your figures contain several parts, the parts should form a neat rectangle when assembled. Choosing the right electronic format at this stage will speed up the processing of your paper and give the best possible results in print. We would like the figures to be supplied as vector files - EPS, PDF, AI or postscript (PS) file formats (not raster or bitmap files),

preferably generated with vector-graphics software (Adobe Illustrator for example). Please try to ensure that all figures are non-flattened and fully editable. All images should be at least 300 dpi resolution (when figures are scaled to approximately the size that they are to be printed at) and in RGB colour format. Please do not submit Jpeg or flattened TIFF files. Please see also 'Guidelines for Electronic Submission of Figures' at the end of this letter for further detail.

Figure legends must provide a brief description of the figure and the symbols used, within 350 words, including definitions of any error bars employed in the figures.

Please include a statement before the acknowledgements naming the author to whom correspondence and requests for materials should be addressed.

Finally, we require authors to include a statement of their individual contributions to the paper -- such as experimental work, project planning, data analysis, etc. -- immediately after the acknowledgements. The statement should be short, and refer to authors by their initials. For details please see the Authorship section of our joint Editorial policies at http://www.nature.com/authors/editorial_policies/authorship.html

- * include a point-by-point response to any editorial suggestions and to our referees. Please include your response to the editorial suggestions in your cover letter, and please upload your response to the referees as a separate document.
- * ensure it complies with our format requirements for Letters as set out in our guide to authors at www.nature.com/nmicrobiol/info/gta/
- * state in a cover note the length of the text, methods and legends; the number of references; number and estimated final size of figures and tables
- * resubmit electronically if possible using the link below to access your home page:

Link Redacted

*This url links to your confidential homepage and associated information about manuscripts you may have submitted or be reviewing for us. If you wish to forward this e-mail to co-authors, please delete this link to your homepage first.

Please ensure that all correspondence is marked with your Nature Microbiology reference number in the subject line.

Nature Microbiology is committed to improving transparency in authorship. As part of our efforts in this direction, we are now requesting that all authors identified as 'corresponding author' on published papers create and link their Open Researcher and Contributor Identifier (ORCID) with their account on the Manuscript Tracking System (MTS), prior to acceptance. This applies to primary research papers only. ORCID helps the scientific community achieve unambiguous attribution of all scholarly contributions. You can create and link your ORCID from the home page of the MTS by clicking on 'Modify my Springer Nature account'. For more information please visit www.springernature.com/orcid.

We hope to receive your revised paper within three weeks. If you cannot send it within this time, please let us know.

Yours sincerely,

Reviewer Expertise:

Referee #1: Virology, Proteome, Structure
Referee #2: Deep mutational scanning
Referee #3: Population genetics, Virus Evolution, Pathogen Genomics

Reviewers Comments:

Reviewer #1 (Remarks to the Author):

Bakhache et al. provide a well-written and detailed revision of their manuscript. One substantial change in this version is the removal of all structure prediction data due to different recommendation of reviewer 2 and myself on the general topic. To evaluate this decision at least two different use cases need to be distinguished for which structural predictions would be useful:

1) As a "scaffold" to map regions of higher InDel tolerance onto proteins with no deposited experimental structure and determine in which secondary structural elements InDels and substitutions are better tolerated as done in Fig. 4, 5, and S8. From my viewpoint, there are no objections in the field right now to using structural predictions in the absence of experimental data for this kind of question as long as the models are well-validated along their scores (pLDDT, PAE, and pTM). An already 1,5 years old community assessment states: "In line with the reported high accuracy of the models, we found that AF2-predicted structures, on average, tend to give results that are as good as those derived from experimental structures." (<https://www.nature.com/articles/s41594-022-00849-w>). I also respectfully disagree with reviewer 2's

conclusion about AlphaFold 2 showing low accuracy in unstructured regions, in case understood his point correctly. Intrinsically disordered regions should have low localization accuracy by definition. Moreover, the presence of IDRs is generally predicted extremely well by AF2. So well that protein disorder databases such as MobiDB rely on AF2 to detect IDRs by their low pLDDT. Also, if a secondary structure has a high pLDDT in a generally well-folded prediction, I do not see any issue in calling this feature to evaluate InDel frequency.

Based on this, I do not see any reason not to use structural predictions for the remaining viral proteins as long as the scores are provided. In my opinion, including them, at least as a reference for future work, would be very useful.

2) To predict the effect of changes in regards to protein stability. Reviewer 2 and the authors are correct that this is a point of some debate in the field. However there is mounting evidence despite some negative reports that AF2 and now AF3 in combination with other methods such as "physics-based" modelling do work in evaluating the effect of sequence changes on protein stability and folding. However, I agree with the authors point that this might be out of the scope of this manuscript.

In conclusion, the authors have put a lot of extra work and effort into analysing the available experimental structures which makes the manuscript much stronger. I would appreciate if the authors would include AF2 or AF3 predictions of the remaining proteins for which no experimental structures are available, also to complete the analysis and provide a reference for future use.

Otherwise I do not have any further objections and would like to congratulate the authors on this very well-designed study and the very well-written paper.

Reviewer #1 (Remarks on code availability):

I downloaded the Dryad depository and explored some of the code. Many of the scripts are well-documented, and the readme file gives a detailed but concise overview of the included data, scripts, and needed dependencies. However, due to time restrictions, I have not run any of the code.

Reviewer #2 (Remarks to the Author):

Fantastic job revising the manuscript. I appreciate all the changes the authors make and believe it to be an impactful manuscript with broad interest!

Willow Coyote-Maestas

Reviewer #3 (Remarks to the Author):

The authors have submitted an extensive revision of their manuscript and that address most points raised in my previous review. I think high throughput methodologies to assess the phenotypic impact of indel variation are important and in this light, this manuscript makes an important contribution. It is nice to see the correspondence between those locations with indel diversity among difference EV-A viruses.

remaining comment:

- the correlation of replicate results for indels in supp fig 5 D&E have some strange off-diagonal lines. Furthermore, why is the correlation more or less exact for some parts of the graphs for InDels, while one gets a more expected scatter for the substitutions?

Version 2:

Reviewer comments:

Reviewer #1

(Remarks to the Author)

I thank the authors for adding the AF3 predictions to the manuscript. I have no further queries and would like to congratulate the authors again on this very thoughtful study.

Reviewer #4

(Remarks to the Author)

Summary:

This work performed InDel mutagenesis of Enterovirus A71 to complement the current literature based on point mutations to understand protein tolerance. They generated libraries with an 8 amino acid insertion or 1, 2, or 3 amino acid deletion. Sequencing of the replication competent viruses identified tolerance hotspots, predominantly in 2A(pro) and VP1. Phylogenetic analysis was able to find evidence for the VP1 hotspot in the natural history of EV-A. This manuscript is clearly written and easy to follow.

Minor:

VP1 is a high copy number capsid protein and has been used in the past as a candidate for the incorporation of tags. How does the hotspot identified in this work compare to the positions used in the literature? Does the screen identify a better position for tagging? See Stirk et al.

<https://doi.org/10.1093/protein/7.1.47> and Wang et al. <https://doi.org/10.1099/vir.0.040253-0>.

Decision Letter:

13th August 2024

Dear Patrick,

Thank you for your patience while your manuscript "A Comprehensive Map of Evolutionary Constraints Across the Enterovirus A Genome" was under peer-review at Nature Microbiology. It has now been seen by 2 referees, whose expertise and comments you will find at the of this email. You will see from their comments below that while they find your work of interest, some important points are raised.

As mentioned before, we need to make sure that your code can be re-run by others, so an additional referee was recruited to evaluate this aspect, as this was not done by the other referees before.

We are very interested in the possibility of publishing your study in Nature Microbiology, but would like to consider your response to these concerns in the form of a revised manuscript before we make a final decision on publication.

In particular, you will see that referee #5 requests additional clarifications on which dependencies there are for the code, as well as to add demo data. These rest referees' reports are clear and the remaining issues should be straightforward to address.

If you have not done so already please begin to revise your manuscript so that it conforms to our Article format instructions at <http://www.nature.com/nmicrobiol/info/final-submission/>

The usual length limit for a Nature Microbiology Article is six display items (figures or tables) and 3,000 words. We have some flexibility, and can allow a revised manuscript at 3,500 words, but please consider this a firm upper limit. There is a trade-off of ~250 words per display item, so if you need more space, you could move a Figure or Table to Supplementary Information.

Some reduction could be achieved by focusing any introductory material and moving it to the start of your opening 'bold' paragraph, whose function is to outline the background to your work, describe in a sentence your new observations, and explain your main conclusions. The discussion should also be limited. Methods should be described in a separate section following the discussion, we do not place a word limit on Methods.

Nature Microbiology titles should give a sense of the main new findings of a manuscript, and should not contain punctuation. Please keep in mind that we strongly discourage active verbs in titles, and that they should ideally fit within 90 characters each (including spaces).

Please include a data availability statement as a separate section after Methods but before references, under the heading "Data Availability". This section should inform readers about the availability of the data used to support the conclusions of your study. This information includes accession codes to public repositories (data banks for protein, DNA or RNA sequences, microarray, proteomics data etc...), references to source data published alongside the paper, unique identifiers such as URLs to data repository entries, or data set DOIs, and any other statement about data availability. At a minimum, you should include the following statement: "The data that support the findings of this study are available from the corresponding author upon request", mentioning any restrictions on availability. If DOIs are provided, we also strongly encourage including these in the Reference list (authors, title, publisher (repository name), identifier, year). For more guidance on how to write this section please see: <http://www.nature.com/authors/policies/data/data-availability-statements-data-citations.pdf>

To improve the accessibility of your paper to readers from other research areas, please pay particular attention to the wording of the paper's opening bold paragraph, which serves both as an introduction and as a brief, non-technical summary in about 150 words. If, however, you require one or two extra sentences to explain your work clearly, please include them even if the paragraph is over-length as a result. The opening paragraph should not contain references. Because scientists from other sub-disciplines will be interested in your results and their implications, it is important to explain essential but specialised terms concisely. We suggest you show your summary paragraph to colleagues in other fields to uncover any problematic concepts.

If your paper is accepted for publication, we will edit your display items electronically so they conform to our house style and will reproduce clearly in print. If necessary, we will re-size figures to fit single or double column width. If your figures contain several parts, the parts should form a neat rectangle when assembled. Choosing the right electronic format at this stage will speed up the processing of your paper and give the best possible results in print. We would like the figures to be supplied as vector files - EPS, PDF, AI or postscript (PS) file formats (not raster or bitmap files), preferably generated with vector-graphics software (Adobe Illustrator for example). Please try to ensure that all figures are non-flattened and fully editable. All images should be at least 300 dpi resolution (when figures are scaled to approximately the size that they are to be printed at) and in RGB colour format. Please do not submit Jpeg or flattened TIFF files. Please see also 'Guidelines for Electronic Submission of Figures' at the end of this letter for further detail.

Figure legends must provide a brief description of the figure and the symbols used, within 350 words, including definitions of any error bars employed in the figures.

When submitting the revised version of your manuscript, please pay close attention to our [href="https://www.nature.com/nature-research/editorial-policies/image-integrity">Digital Image Integrity Guidelines.](https://www.nature.com/nature-research/editorial-policies/image-integrity) and to the following points below:

Please include a statement before the acknowledgements naming the author to whom correspondence and requests for materials should be addressed.

Finally, we require authors to include a statement of their individual contributions to the paper -- such as experimental work, project planning, data analysis, etc. -- immediately after the acknowledgements. The statement should be short, and refer to authors by their initials. For details please see the Authorship section of our joint Editorial policies at http://www.nature.com/authors/editorial_policies/authorship.html

* include a point-by-point response to any editorial suggestions and to our referees. Please include your response to the editorial suggestions in your cover letter, and please upload your response to the referees as a separate document.

* ensure it complies with our format requirements for Letters as set out in our guide to authors at www.nature.com/nmicrobiol/info/gta/

* state in a cover note the length of the text, methods and legends; the number of references; number and estimated final size of figures and tables

* resubmit electronically if possible using the link below to access your home page:

Link Redacted

*This url links to your confidential homepage and associated information about manuscripts you may have submitted or be reviewing for us. If you wish to forward this e-mail to co-authors, please delete this link to your homepage first.

Please ensure that all correspondence is marked with your Nature Microbiology reference number in the subject line.

Nature Microbiology is committed to improving transparency in authorship. As part of our efforts in this direction, we are now requesting that all authors identified as 'corresponding author' on published papers create and link their Open Researcher and Contributor Identifier (ORCID) with their account on the Manuscript Tracking System (MTS), prior to acceptance. This applies to primary research papers only. ORCID helps the scientific community achieve unambiguous attribution of all scholarly contributions. You can create and link your ORCID from the home page of the MTS by clicking on 'Modify my Springer Nature account'. For more information please visit [please visit www.springernature.com/orcid](http://www.springernature.com/orcid).

We hope to receive your revised paper within three weeks. If you cannot send it within this time, please let us know.

Yours sincerely,

Reviewer Expertise:

Referee #1: Virology, Proteome, Structure

Referee #4: Virology, Code

Reviewers Comments:

Reviewer #1 (Remarks to the Author):

I thank the authors for adding the AF3 predictions to the manuscript. I have no further queries and would like to congratulate the authors again on this very thoughtful study.

Reviewer #4 (Remarks to the Author):

Summary:

This work performed InDel mutagenesis of Enterovirus A71 to complement the current literature based on point mutations to understand protein tolerance. They generated libraries with an 8 amino acid insertion or 1, 2, or 3 amino acid deletion. Sequencing of the replication competent viruses identified tolerance hotspots, predominantly in 2A(pro) and VP1. Phylogenetic analysis was able to find evidence for the VP1 hotspot in the natural history of EV-A. This manuscript is clearly written and easy to follow.

Minor:

VP1 is a high copy number capsid protein and has been used in the past as a candidate for the incorporation of tags. How does the hotspot identified in this work compare to the positions used in the literature? Does the screen identify a better position for tagging? See Stirk et al. <https://doi.org/10.1093/protein/7.1.47> and Wang et al. <https://doi.org/10.1099/vir.0.040253-0>.

Reviewer #4 (Remarks on code availability):

Summary:

The authors provide numerous data files including their python and R code, and they deposited the sequencing runs at NCBI. The included readme files list the directory tree as well as descriptions of the included files. However, I was unable to find demo data to test the code with, there are no running instructions, and the python dependencies were not listed. Demo data would not only facilitate the review process but also be helpful as a positive test of the pipeline for other labs that are interested in this technique.

Major

1. Demo data should be included with the python and R code. The description of the python scripts states that the input for both is a mapped .sam file. A small .sam file for testing in a reasonable amount of time should be included, along with a test file for the "query" and "templateFasta" arguments. It is unclear what the input files for the R scripts are.

2. The authors should include running instructions. The R code returns numerous "No such file or directory" errors. Instructions should state what files the code is dependent on and the directory tree, e.g. should all of the files be in a single folder and the R working directory should be set to that folder. The instructions should also state that the python code should be run from the command line and include the command syntax. While one can

deduce that from reading the .py file, it would be helpful if it was explicitly stated in the instructions with the arguments for calling the demo data. The instructions should also state what the input files are and what the output files should look like.

3. The dependencies/packages should be completely listed. For python, the dependencies should be listed and include the version numbers. The R packages for Data_Generation_Markdown_InDel_Manuscript.r are listed in the readme files, but the packages for EVA71_EntropyAnalysis.R are missing.

Minor

1. There are 2 readme files: Dryad_Repo_InDel_Readme.txt and README.md. These appear to be mostly redundant and should be consolidated. I like the directory tree in README.md and found it helpful.

2. In stickleback.0.2.py, should line 43 be <4? There are 3 arguments (pathto/input.sam, query, templateFasta) and python counts the file.py argument as arg[0]. Therefore, if the right number of arguments are entered, it should be 4 (stickleback.py, pathto/input.sam, query, templateFasta).
if len(args)<4:
instead of
if len(args)<3:

Entering the command:

```
python .\stickleback.0.2.py input.sam query
```

Lead to the following error instead of the check that would state the usage:

```
template=args[3]
```

```
IndexError: list index out of range
```

3. stickleback.0.2.py imports "re" but doesn't use it.

4. DelMapper_v0.2.py imports "pandas" but doesn't use it.

5. In DelMapper_v0.2.py, it looks like each function has a description commented above it. However, the function "parselist" has "posParse" written in the comment and the comment is the same for all 3 functions.

6. It would be nice if DelMapper_v0.2.py also included a syntax check and return the usage when an incorrect number of arguments are used, like stickleback.0.2.py.

Version 3:

Reviewer comments:

Reviewer #4

(Remarks to the Author)

The github pages are very well organized with clear instructions.

Testing:

stickleback.py: runs as explained

smelt.py example and usage are incorrect

The example command is missing the template.fasta argument.

The usage is missing the output.csv argument.

In addition, it requires the outdated numpy v1.x

EVA71_EntropyAnalysis.R: runs as explained

DMS_Processing_Workflow.R: runs as explained in RStudio but not R.

In R, every call to the codonFilter() function generates the following error:

```
> capsid_PO_filtered <- codonFilter(oligos_programmed_capsid, "input_files/codon_counts/Capsid_input_DMS.codonCounts", wildtype_capsid, start_end_capsid)
```

```
Rows: 81889 Columns: 1
```

```
— Column
```

```
specification—
```

```
Delimiter: "\t"
```

```
chr (1): >Capsid_DMS-1_Met2Cys
```

i Use `spec()` to retrieve the full column specification for this data.

i Specify the column types or set `show_col_types = FALSE` to quiet this message.

```
Rows: 2194 Columns: 67
```

```
— Column
```

```
specification—
```

```
Delimiter: "\t"
```

```
dbl (67): AAA, AAC, AAG, AAT, ACA, ACC, ACG, ACT, AGA, AGC, AGG, AGT, ATA, ATC, ATG, ATT, CAA, CAC, CAG, CAT, CCA, CCC, CCG, CCT, CGA...
```

i Use `spec()` to retrieve the full column specification for this data.

i Specify the column types or set `show_col_types = FALSE` to quiet this message.

```
Error in h(simpleError(msg, call)) :
```

error in evaluating the argument 'x' in selecting a method for function 'slice': Rle of type 'list' is not supported

Major:

1. Please state the version of the python dependencies. As outlined above, this is important for smelt.py
2. Correct the smelt.py example and usage information. Make it clear that it uses an outdated numpy version or update the code to use the current version of numpy.
3. Check if you can reproduce the DMS_Processing_Workflow.R error in a fresh install of R. If so, please make a comment on github.

Reviewer #5

(Remarks to the Author)

Overall, this is a good study. It provides a wealth of information enterovirus A, and also is the most comprehensive study to date on insertions versus deletions versus substitutions to a viral protein.

Minor comments:

- In Figure 1A-F, one surprising aspect is that some deletions and insertions (as well as substitutions) have extremely positive scores, suggesting they improve growth a lot. Is this a real result or experimental noise? Ideally this could be validated for some key mutations, or at least discussed.

- In the text, a bit more discussion should be given to the fact that the insertional mutants are all of a specific length and context (glycine-serine flanked). It is possible that the effects of insertions would be different if they were shorter, etc. I don't expect the authors to test every possible insertion length/type (which are basically infinite), but they should more clearly explain they are testing a specific length and context.

- Line 47-48: A few recent viral protein DMS experiments have included some deletions. Those studies don't have insertions and are not nearly as comprehensive as this one, so they do not in any way decrease the novelty and impact of the current study. However, for completeness they should be cited, eg: deletions in SARS-CoV-2 RBD (<https://journals.plos.org/plospathogens/article?id=10.1371/journal.ppat.1011901>), deletions in some sites in SARS-CoV-2 spike including NTD (<https://www.nature.com/articles/s41586-024-07636-1>)

Decision Letter:

4th September 2024

Dear Patrick,

Thank you for your patience and your understanding, while we recruited an additional reviewer and while your manuscript "A Comprehensive Map of Evolutionary Constraints Across the Enterovirus A Genome" was under peer-review at Nature Microbiology. It has now been seen by 2 referees (one previously looking at your code and an additional one to assess the technical aspects of the mutational scanning, as mentioned before). You will find their comments the bottom of this email. You will see from their comments below that while they find your work of interest, some important points are raised. We are very interested in the possibility of publishing your study in Nature Microbiology, but would like to consider your response to these concerns in the form of a revised manuscript before we make a final decision on publication.

In particular, you will see that referee #4, still raises issues with your code resulting in errors and suggests to:

1. State the version of the python dependencies.
2. Correct the smelt.py example and usage information. Make it clear that it uses an outdated numpy version or update the code to use the current version of numpy.
3. Check if you can reproduce the DMS_Processing_Workflow.R error in a fresh install of R. If so, please make a comment on github.

The rest referees' reports are clear and the remaining issues should be straightforward to address.

I am sorry for the additional round of revision but we have to make sure that the code can be used by others.

If you have not done so already please begin to revise your manuscript so that it conforms to our Article format instructions at <http://www.nature.com/nmicrobiol/info/final-submission/>

The usual length limit for a Nature Microbiology Article is six display items (figures or tables) and 3,000 words. We have some flexibility, and can allow a revised manuscript at 3,500 words, but please consider this a firm upper limit. There is a trade-off of ~250 words per display item, so if you need more space, you could move a Figure or Table to Supplementary Information.

Some reduction could be achieved by focusing any introductory material and moving it to the start of your opening 'bold' paragraph, whose function is to outline the background to your work, describe in a sentence your new observations, and explain your main conclusions. The discussion should also be limited. Methods should be described in a separate section following the discussion, we do not place a word limit on Methods.

Nature Microbiology titles should give a sense of the main new findings of a manuscript, and should not contain punctuation. Please keep in mind that we strongly discourage active verbs in titles, and that they should ideally fit within 90 characters each (including spaces).

Please include a data availability statement as a separate section after Methods but before references, under the heading "Data Availability". This section should inform readers about the availability of the data used to support the conclusions of your study. This information includes accession codes to public repositories (data banks for protein, DNA or RNA sequences, microarray, proteomics data etc...), references to source data published alongside the paper, unique identifiers such as URLs to data repository entries, or data set DOIs, and any other statement about data availability. At a minimum, you should include the following statement: "The data that support the findings of this study are available from the corresponding author

upon request", mentioning any restrictions on availability. If DOIs are provided, we also strongly encourage including these in the Reference list (authors, title, publisher (repository name), identifier, year). For more guidance on how to write this section please see: <http://www.nature.com/authors/policies/data/data-availability-statements-data-citations.pdf>

To improve the accessibility of your paper to readers from other research areas, please pay particular attention to the wording of the paper's opening bold paragraph, which serves both as an introduction and as a brief, non-technical summary in about 150 words. If, however, you require one or two extra sentences to explain your work clearly, please include them even if the paragraph is over-length as a result. The opening paragraph should not contain references. Because scientists from other sub-disciplines will be interested in your results and their implications, it is important to explain essential but specialised terms concisely. We suggest you show your summary paragraph to colleagues in other fields to uncover any problematic concepts.

If your paper is accepted for publication, we will edit your display items electronically so they conform to our house style and will reproduce clearly in print. If necessary, we will re-size figures to fit single or double column width. If your figures contain several parts, the parts should form a neat rectangle when assembled. Choosing the right electronic format at this stage will speed up the processing of your paper and give the best possible results in print. We would like the figures to be supplied as vector files - EPS, PDF, AI or postscript (PS) file formats (not raster or bitmap files), preferably generated with vector-graphics software (Adobe Illustrator for example). Please try to ensure that all figures are non-flattened and fully editable. All images should be at least 300 dpi resolution (when figures are scaled to approximately the size that they are to be printed at) and in RGB colour format. Please do not submit Jpeg or flattened TIFF files. Please see also 'Guidelines for Electronic Submission of Figures' at the end of this letter for further detail.

Figure legends must provide a brief description of the figure and the symbols used, within 350 words, including definitions of any error bars employed in the figures.

When submitting the revised version of your manuscript, please pay close attention to our [href="https://www.nature.com/nature-research/editorial-policies/image-integrity">Digital Image Integrity Guidelines.](https://www.nature.com/nature-research/editorial-policies/image-integrity) and to the following points below:

Please include a statement before the acknowledgements naming the author to whom correspondence and requests for materials should be addressed.

Finally, we require authors to include a statement of their individual contributions to the paper -- such as experimental work, project planning, data analysis, etc. -- immediately after the acknowledgements. The statement should be short, and refer to authors by their initials. For details please see the Authorship section of our joint Editorial policies at http://www.nature.com/authors/editorial_policies/authorship.html

- * include a point-by-point response to any editorial suggestions and to our referees. Please include your response to the editorial suggestions in your cover letter, and please upload your response to the referees as a separate document.
- * ensure it complies with our format requirements for Letters as set out in our guide to authors at www.nature.com/nmicrobiol/info/gta/
- * state in a cover note the length of the text, methods and legends; the number of references; number and estimated final size of figures and tables
- * resubmit electronically if possible using the link below to access your home page:

Link Redacted

*This url links to your confidential homepage and associated information about manuscripts you may have submitted or be reviewing for us. If you wish to forward this e-mail to co-authors, please delete this link to your homepage first.

Please ensure that all correspondence is marked with your Nature Microbiology reference number in the subject line.

Nature Microbiology is committed to improving transparency in authorship. As part of our efforts in this direction, we are now requesting that all authors identified as 'corresponding author' on published papers create and link their Open Researcher and Contributor Identifier (ORCID) with their account on the Manuscript Tracking System (MTS), prior to acceptance. This applies to primary research papers only. ORCID helps the scientific community achieve unambiguous attribution of all scholarly contributions. You can create and link your ORCID from the home page of the MTS by clicking on 'Modify my Springer Nature account'. For more information please visit [please visit www.springernature.com/orcid](http://www.springernature.com/orcid).

We hope to receive your revised paper within three weeks. If you cannot send it within this time, please let us know.

Yours sincerely,

Reviewer Expertise:

Referee #4: Code

Referee #5: Deep mutational scanning

Reviewers Comments:

Reviewer #4 (Remarks to the Author):

The github pages are very well organized with clear instructions.

Testing:

stickleback.py: runs as explained

smelt.py example and usage are incorrect
The example command is missing the template.fasta argument.
The usage is missing the output.csv argument.
In addition, it requires the outdated numpy v1.x

EVA71_EntropyAnalysis.R: runs as explained

DMS_Processing_Workflow.R: runs as explained in RStudio but not R.
In R, every call to the codonFilter() function generates the following error:
> capsid_P0_filtered <- codonFilter(oligos_programmed_capsid, "input_files/codon_counts/Capsid_input_DMS.codonCounts", wildtype_capsid, start_end_capsid)
Rows: 81889 Columns: 1
—— Column
specification
Delimiter: "\t"
chr (1): >Capsid_DMS-1_Met2Cys

i Use `spec()` to retrieve the full column specification for this data.
i Specify the column types or set `show_col_types = FALSE` to quiet this message.
Rows: 2194 Columns: 67

—— Column
specification
Delimiter: "\t"
dbl (67): AAA, AAC, AAG, AAT, ACA, ACC, ACG, ACT, AGA, AGC, AGG, AGT, ATA, ATC, ATG, ATT, CAA, CAC, CAG, CAT, CCA, CCC, CCG, CCT, CGA...

i Use `spec()` to retrieve the full column specification for this data.
i Specify the column types or set `show_col_types = FALSE` to quiet this message.
Error in h(simpleError(msg, call)) :
error in evaluating the argument 'x' in selecting a method for function 'slice': Rle of type 'list' is not supported

Major:

1. Please state the version of the python dependencies. As outlined above, this is important for smelt.py
2. Correct the smelt.py example and usage information. Make it clear that it uses an outdated numpy version or update the code to use the current version of numpy.
3. Check if you can reproduce the DMS_Processing_Workflow.R error in a fresh install of R. If so, please make a comment on github.

Reviewer #5 (Remarks to the Author):

Overall, this is a good study. It provides a wealth of information enterovirus A, and also is the most comprehensive study to date on insertions versus deletions versus substitutions to a viral protein.

Minor comments:

- In Figure 1A-F, one surprising aspect is that some deletions and insertions (as well as substitutions) have extremely positive scores, suggesting they improve growth a lot. Is this a real result or experimental noise? Ideally this could be validated for some key mutations, or at least discussed.

- In the text, a bit more discussion should be given to the fact that the insertional mutants are all of a specific length and context (glycine-serine flanked). It is possible that the effects of insertions would be different if they were shorter, etc. I don't expect the authors to test every possible insertion length/type (which are basically infinite), but they should more clearly explain they are testing a specific length and context.

- Line 47-48: A few recent viral protein DMS experiments have included some deletions. Those studies don't have insertions and are not nearly as comprehensive as this one, so they do not in any way decrease the novelty and impact of the current study. However, for completeness they should be cited, eg: deletions in SARS-CoV-2 RBD (<https://journals.plos.org/plospathogens/article?id=10.1371/journal.ppat.1011901>), deletions in some sites in SARS-CoV-2 spike including NTD (<https://www.nature.com/articles/s41586-024-07636-1>)

Version 4:

Decision Letter:

Our ref: NMICROBIOL-24010043D

16th September 2024

Dear Dr. Dolan,

Thank you for submitting your revised manuscript "A Comprehensive Map of Evolutionary Constraints Across the Enterovirus A Genome" (NMICROBIOL-24010043D). It has now been seen by the original referees and their comments are below. The reviewers find that the paper has improved in revision, and therefore we'll be happy in principle to publish it in Nature Microbiology, pending minor revisions to satisfy the referees' final requests and to comply with our editorial and formatting guidelines.

Thank you again for your interest in Nature Microbiology Please do not hesitate to contact me if you have any questions.

Sincerely,

Version 5:

Decision Letter:

24th October 2024

Dear Patrick,

I am delighted to accept your Article "Deep mutation, insertion, and deletion scanning across the Enterovirus A proteome reveals constraints shaping viral evolution" for publication in Nature Microbiology. Thank you for having chosen to submit your work to us and many congratulations.

Please note that *Nature Microbiology* is a Transformative Journal (TJ). Authors may publish their research with us through the traditional subscription access route or make their paper immediately open access through payment of an article-processing charge (APC). Authors will not be required to make a final decision about access to their article until it has been accepted. <[a href="https://www.springernature.com/gp/open-research/transformative-journals"](https://www.springernature.com/gp/open-research/transformative-journals)> Find out more about Transformative Journals

Authors may need to take specific actions to achieve <[a href="https://www.springernature.com/gp/open-research/funding/policy-compliance-faqs"](https://www.springernature.com/gp/open-research/funding/policy-compliance-faqs)> compliance with funder and institutional open access mandates. If your research is supported by a funder that requires immediate open access (e.g. according to <[a href="https://www.springernature.com/gp/open-research/plan-s-compliance"](https://www.springernature.com/gp/open-research/plan-s-compliance)>Plan S principles) then you should select the gold OA route, and we will direct you to the compliant route where possible. For authors selecting the subscription publication route, the journal's standard licensing terms will need to be accepted, including <[a href="https://www.nature.com/nature-portfolio/editorial-policies/self-archiving-and-license-to-publish"](https://www.nature.com/nature-portfolio/editorial-policies/self-archiving-and-license-to-publish)>self-archiving policies. Those licensing terms will supersede any other terms that the author or any third party may assert apply to any version of the manuscript.

An online order form for reprints of your paper is available at <[a href="https://www.nature.com/reprints/author-reprints.html"](https://www.nature.com/reprints/author-reprints.html)><https://www.nature.com/reprints/author-reprints.html>. All co-authors, authors' institutions and authors' funding agencies can order reprints using the form appropriate to their geographical region.

We welcome the submission of potential cover material (including a short caption of around 40 words) related to your manuscript; suggestions should

be sent to Nature Microbiology as electronic files (the image should be 300 dpi at 210 x 297 mm in either TIFF or JPEG format). Please note that such pictures should be selected more for their aesthetic appeal than for their scientific content, and that colour images work better than black and white or grayscale images. Please do not try to design a cover with the Nature Microbiology logo etc., and please do not submit composites of images related to your work. I am sure you will understand that we cannot make any promise as to whether any of your suggestions might be selected for the cover of the journal.

Congrats again to you and your co-authors! I am looking forward to seeing your paper published.

With kind regards,

P.S. Click on the following link if you would like to recommend Nature Microbiology to your librarian
<http://www.nature.com/subscriptions/recommend.html#forms>

** Visit the Springer Nature Editorial and Publishing website at http://editorial-jobs.springernature.com?utm_source=ejP_NMicro_email&utm_medium=ejP_NMicro_email&utm_campaign=ejp_NMicro for more information about our career opportunities. If you have any questions please click [here](mailto:editorial.publishing.jobs@springernature.com). **

Reviewer #1 (Remarks to the Author): Virology, Proteome, Structure

Summary:

The manuscript by Bakhache et al. describes the results of a large InDel screen on Enterovirus A71. The authors find that defined regions in the viral proteome are more prone to tolerate insertions or deletions of amino acids with hotspots in the viral protease 2A(pro) and in parts of the capsid protein VP1. These regions match with evolutionary less conserved sequence spaces. The authors go on to map the regions that tolerate InDels to available structural data and also try to predict the effect of two InDels on protein structure using AlphaFold.

The manuscript is very well-written, concise, and easily understandable. In the current version, however, the authors do only limited analysis of the potential structural effects of these InDels which is not sufficient.

Major concerns:

-The analysis depicted in Fig. 4, is currently very limited as only one protein (2A(pro)) is analyzed. This analysis of which structural features tolerate InDels should be done for all A71 proteins to make the conclusions significant. If experimental structures do not exist, AlphaFold predictions can be used if their scores (MSA depth, pLDDT, PAE) indicate well-predicted structures. (Please refer to the original publication by Jumper et al. and the Deepmind/EMBL Alphafold database FAQ for guidance if needed.) Using AlphaFold predictions for all A71 proteins would also have the benefit that the pLDDT values would indicate flexible disordered regions (pLDDT below 50), and the PAE data would indicate structural flexibility between domains. These areas should be enriched for InDel tolerance according to the conclusions of this manuscript. Please also see my last comment below on Fig 5M-O in this regard.

We thank the reviewer for this suggestion. To provide a more thorough interpretation, we have now performed structural analysis on all viral proteins with available structures, excluding the two smallest viral proteins 2B and 3B. These analyses can be found in the updated Figures 4, 5, and Supplementary Figure 8. We have also included the same analyses for the newly added amino acid substitution scanning data that we have incorporated throughout the manuscript. For all the structures, we observed an overlap between tolerance for InDels and amino acid substitutions. Finally, to address aspects of the structural impacts of mutation, we analyzed the tolerance to amino acids with different propensities to form alpha helices (Figure 4 D and K).

-Based on the current analysis, it is unclear why most reported InDels are not tolerated. Instead of providing few examples, the authors should provide a more exhaustive

analysis of the likely effect on protein stability (e.g. $\Delta\Delta G$ stability changes using Rosetta, see <https://www.nature.com/articles/s41594-022-00849-w>), active sites and protein-protein interactions (e.g. using AlphaFold-Multimer v2.3 and scoring by ipTM).

Through deep population sequencing, we have described that InDels in Enterovirus populations are mostly lethal to viral growth (<https://www.pnas.org/doi/full/10.1073/pnas.2304667120>). This corresponds with the experimentally measured InDel tolerance in our screen. The lethality of InDels relative to AA substitutions is likely due to their two-hit fitness effects, first changing the residues at a given site, and second changing the length of structural features and eliciting more global structural distortions. A comprehensive computational biophysical analysis would be valuable, however such a study falls outside of our intended scope, and additionally outside of our expertise.

As suggested by reviewer 2, several important caveats should be considered when using such approaches, and we have therefore focused on interpretations based on published structures. As an example, the ability of the 2A protein to tolerate InDels at many locations might reflect its minor role in the replication complex or reflects functional promiscuity as a protease targeting multiple host factors which we describe in Fig. 4B. We have also added an analysis of the 3A protein dimer in interaction with its host factor ACBD3 that shows mutational constraints at sites of interaction and dimer interfaces (Supplementary figure 8, A-C). In addition, in Figure 5, we show the tolerance to InDels in the assembled viral capsid are localized to unstructured regions that are not at the protomer-protomer interfaces.

-Fig 5M-O: From the provided data (the linked Dryad repository wasn't working for me), it is unclear how good the VP1 prediction is. The authors should provide MSA, pLDDT, PAE and pTM data for reference. Moreover, whether VP1 was predicted as a monomer or pentamer is unclear. To be able to compare to the pentamer structure in 5A-L, the predictions would need to be run as a pentamer (e.g., using local ColabFold in Multimer mode and enough GPU RAM and enough cycles until the model pLDDTs converge). This might also change the position of the EF loop, which is placed differently in the current prediction (compare Fig 5I-L with 5 M-O).

Our apologies for the broken link to dryad. We have now updated the reviewer link and confirmed it works.

-Fig 5M-O: to evaluate the utility of AlphaFold predictions to detect the effect of InDels, the authors need to predict at least a representative number of structures with deleterious and tolerated InDels and quantify their effect not only on local structure but also on global protein folding either using approaches such as those described here: <https://journals.aps.org/prl/abstract/10.1103/PhysRevLett.131.218401> or the link provided above.

Finally, when evaluating loop positions, as shown in Fig 5M-O, the PAE and pLDDT values need to be plotted to ensure that the predicted loop position is not ambiguous due to low scores.

While we are eager to use these tools to explore the effects of mutation, the field, as made plain through these reviews, still disagrees on the validity of interpreting the effects of InDels on structures generated by these tools. We appreciate the guidance from the reviewer on doing this most rigorously, however, due to the conflicting views of the reviewers, we have removed the AlphaFold2 predictions to ensure we aren't over interpreting our results. We are currently exploring available pipelines for their appropriateness for evaluating mutational fitness effects and will implement your guidance in those subsequent analyses.

Minor:

The last sentence of the second paragraph in the introduction is not grammatically correct.

Thank you for your close reading of the manuscript. Grammatical errors have been fixed.

Reviewer #2 (Remarks to the Author): Deep mutational scanning

This manuscript by Bakhache, et al employs deep InDel scanning on the Enterovirus A71 capsid protein to explore the structural tolerance to insertion and deletions mutations. InDels allow massive jumps in sequence space, but they are unexplored experimentally due to technical challenges associated with making InDels. The authors used recently developed pipelines that solve these problems, SPINE and DIMPLE, that allow systematic efficient assembly of InDel libraries. Taken together, the study is amongst the first systematic surveys of insertions and deletions on a protein and currently (to my knowledge) is the largest protein such approaches have been taken. The authors do an excellent job of comparing the results of the variants to the structural properties of the viral capsid protein. The authors did not have sufficient technical details to follow how the screens were conducted and how data was processed. Additionally, it would be nice to see further discussion of the results of the study with prior studies. Due to the novelties of the studies, yet necessary changes to improve the manuscript, I recommend a revision at Nature Microbiology.

Major Concerns

1. While this study is highly innovative in that it conducts an indel scan on a massive protein. While it would have been nice to see missense variants alongside indels to aid in interpretation, the results here could be contextualized with those in prior studies solely focused on indels. The manuscript will likely be of interest to others who work on indels and as such it could be good to discuss the tolerance seen in this manuscript contextualized with other studies and what secondary structural elements are

differentially sensitive to insertions and deletions. Some manuscripts the authors may want to cite and discuss include a landmark AAV engineering manuscript generated by oligo pools (Ogden, 2019, Science), antibiotic resistance protein (Gonzalez CE, 2019), a recent domain insert on AAV, an indel scan within the beta amyloid protein (Seuma, Nature Comm, 2023), a systematic insertional studies in a potassium channel that included many more inserted motifs using this platform (Coyote-Maestas, Nature Comm, 2022), and a recent preprint that conducted mutational scans across many unrelated proteins (Topolska, biorxiv, 2023).

Using the same pipeline described for InDels, we have now measured the effects of all missense variants in the full viral proteome and have integrated this data into all of our analyses throughout the paper. This has provided some interesting observations where a significant overlap was found in regions that tolerate InDels and amino acid changes. For example, the BC loop of VP1 which tolerates InDels and harbors a highly enriched mutant in the DMS screen. We thank the reviewer for this insightful feedback that has improved this manuscript dramatically. We have also integrated the citations into the introduction and discussion sections.

2. The authors use Alphafold2 to predict a structural model of VP1 (8E2X) capsid proteins including deletions (Fig 5M-Q). While I find it highly interesting to explore the structural consequences of deletions I would urge major caution in interpreting or including this model in the manuscript. From my reading this model is not essential to the manuscript and indel effects on protein structures are under-explored by structural biologists. Alphafold2 was trained on the protein data bank and multiple sequence alignments, which both don't typically handle indels very much. Evidence of indels not being well-handled by Alphafold2 are that the unstructured regions of proteins have low accuracy of prediction in Alphafold2 and the associated variant effect predictor Alpha missense cannot predict indels (Cheng J, 2023, Science). More broadly, Alphafold2 struggles to predict the effects of mutations alone (Pak MA, et al PLoS One, 2023). If the authors do want a structural model, something that incorporates physics would be worthwhile such as molecular dynamics simulations after a perturbation is made. For example, one can include MD simulations after an Alphafold2-like structure prediction effort to relax the ensemble (as in Yee SW, Macdonald CB, 2023, Biorxiv), however that has not been validated well for deletions. My suggestion is to not include these models as they are not essential to the manuscript and do not add much beyond simply mapping the indel scores while implying incorrectly that structural models of indels exist. We have removed the AlphaFold2 predictions from the manuscript. We agree with the reviewer that performing such analysis would be highly interesting in the future once models that are trained on InDels are developed.

3. The methods section of the manuscript is missing many details that are necessary to follow what was done in the screens and what the computational pipelines that were used for analyzing the data. The screen is mentioned in the 'Virus Passaging and Sequencing' section however it is unclear how many passages were done and how long those passages were. Perhaps adding more detail in the workflow cartoon in Figure 2B would be helpful for understanding the experiments as well. Similarly, the fitness scores are currently not mentioned within the method section with the description of data processing ends after alignment prior to going to structural analysis. It is difficult to judge the technical aspects of the indel scanning work and others would likely need more details if they wanted to conduct similar experiments.

We have updated Figure 1B and the methods section with more details on how the screen was performed and the analysis pipelines used. In revision we decided to adopt the widely used enrich2 software for calculating all enrichment scores. This should be sufficient to allow other researchers to conduct similar screens in different viral systems.

Minor Concerns

1. Please add line numbers they help with reviewing and making specific suggestions.

The line numbers have been added to the manuscript.

2. The authors mention that the libraries vary by a modest three and six fold range in insertions and deletions. Was there any systematic pattern to variance? Were shorter vs longer sequences differentially selected? Including this information would be useful for other people who are conducting indel scanning and for improvement for library generation pipelines to reduce bias.

In Supplementary Figure 2, we added boxplots and scatterplots comparing libraries of different deletion lengths. There was no observed bias in libraries with different deletion sizes. Most of the variance obtained in our libraries appears to be related to our ability to pool the sublibraries at equimolar concentrations.

3. From my reading the authors conducted two separate screens for insertions and deletions. They do well to only make comparisons qualitatively across these groups. However, in Figure 2A these are put on the same plot which is misleading. I recommend separating these.

Thank you, we agree and have separated the plots in Figure 2A.

4. Figure 2A- The insertions side of the plot is missing a dotted line as 'neutral insertions' that the neutral deletion has'.

We removed the 'neutral' dotted lines, both to clarify that these are not necessarily "neutral", but have average fitness effects relative to mutations of the same type.

5. The authors do an interesting experiment where they insert highly variable 5-amino acid peptides. Why was a 5 amino acid peptide used? A bit more rationale here could be useful. From my reading the peptide was of a single amino acid type, however the wording in 'The inserted sequence consisted of each of the 20 amino acids flanked by flexible linkers' is a bit confusing. Specifically, what flexible linkers?

We adopted the flexible linkers because of the limitation of the insertional handle for having constant restriction sites. Therefore, we decided to insert a variable amino acid flanked with the same flexible linkers as a way to measure the effects of different amino acids on insertional tolerance. We have added in the text clarifying statements to explain better the rationale behind our decision.

Here are the updated texts in quotes:

Line 257-263: *"The insertional handle was designed such that a single insertional library can be used to generate a library of any sequence of interest at every position by golden gate cloning. We took advantage of this modularity to assess how the composition of an inserted sequence affects tolerance. The sequence of interest was designed with flexible linkers, which enables compatibility with the insertional handle, maintaining a constant restriction cut site. Therefore, the inserted residue consisted of Serine-Glycine-X-Glycine-Serine, where X is every possible amino acid."*

6. In figure 3 D-G the authors compare the physicochemical properties of the peptides within specific regions of the proteins using stacked bar plots. The authors' descriptions in the text are hard to follow as they do not refer to specific regions based on amino acid sequences. From positions ~565-575 Y seem to dominate whereas positions 585-595 K and N seem important. Some more context and discussion here could be useful. It may also be useful to plot just these subregions as heatmaps which would allow easier interpretation. Could also be useful highlighting these regions in Figure 4.

Thank you for this feedback. We have exchanged the stacked bar plots for heat maps for easier visualization and interpretation of the data (Figure 3 C-F), and changed the text accordingly (Line 266-273). We agree that there is tolerance for tyrosines and lysines on VP3 and the N-terminus of VP1 respectively is notable. We have added more context in the discussion (Lines 439-449). This observation could be related to post-translational modifications that can occur at these residues. For example, VP3 of the foot-and-mouth disease virus (FMDV) belonging to the picornaviridae family has

been shown to interact with JAK1/2 and inhibit tyrosine phosphorylation of STAT1 (<https://www.ncbi.nlm.nih.gov/pmc/articles/PMC4845950/>). The interaction with JAK1/2 could lead to phosphorylation of VP3 at these tyrosine residues. With respect to the N-terminus of VP1, lysines have been previously suggested to favor interactions with the negatively charged RNA during assembly (<https://www.nature.com/articles/s41467-023-43029-0>). This might explain why insertions with lysines (and also amino acid changes to lysines in the DMS data) at the N-terminus of VP1 are favored.

7. Figure 5F-H shows relative fitness mapped onto the external surface of the capsid. In this it's abundantly clear from figs 5F and 5H that there are hotspots that have positive fitness for deletions 1 and 3 amino acids long. However, the subpanel of 2AA deletions Fig 5G contains no visible hotspots. Are deletions 1 or 3 Amino acids better tolerated than 2 Amino acid deletions? I find that surprising - why might this be?

Great point; we have done further analysis to clarify the tolerance for different deletion lengths. We find that 1 AA deletions are much more tolerant than 2 and 3 AA deletions (See Figure 3G). However, the region at the BC loop is an exception where it seems to tolerate better 1 AA or 3 AA deletions. This region has been shown to be implicated with Heparan sulfate binding, which is one of the attachment factors of the virus (<https://www.ncbi.nlm.nih.gov/pmc/articles/PMC6093697/>). We are conducting studies now to test if these different deletions affect heparan sulfate interactions.

8. Figure 5I-L shows many subpanels to represent the effects of indels across the structure of the virus. I find it difficult to see exactly what the authors are discussing in the text. While there are really interesting observed trends such as, 'Another interesting observation centered around two antiparallel Beta-strands, with B-strand e and f tolerating a 2 AA and 3 AA deletion.' While if I squint I can see the deletions 'E and F' there are many other motifs shown. For example two aromatics are shown in stick mode as is a substrate or small molecule of sorts which likely is not necessary for the interpretation. Adjusting the camera shading in chimera or pymol can reduce the noise. We have improved the visualization of our chimera rendered images and simplified them to remove unnecessary structures with stick mode.

Willow Coyote-Maestas

Reviewer #3 (Remarks to the Author): Virus evolution

Bakhache and colleagues present an innovative method to probe the effects of small insertions and deletions across the single polypeptide of EV-A 71. Extending DMS methods beyond amino acid substitutions is an exciting advance.

However, I am left a little unsure what exactly I have learned. This is partly due to the fact that the space of InDel variation is vastly bigger than that of substitutions and it is thus harder to interpret the observations. But I also struggle with getting a sense of how accurate the measurements are, what it means for an insertion to be very beneficial etc. I list a few specific points that I would like to see clarified or addressed. In particular, I think additional validation of the experimental assay would make the paper a lot stronger.

We thank the reviewer for their critical feedback. We have reorganized the figures to clarify the importance of our findings. We also modified the manuscript by adding the effects of all missense variants in the full viral proteome to aid in the interpretation of our data, and provide a more complete story.

As further validation of the study, we have now performed an analysis of a subset of variants comparing their replicative fitness to that of WT through direct competition in a pooled screen (Supp. Fig. 7).

Insertion tolerance: I am reading Fig 2A correctly that none of the insertions to VP2 that seem tolerated or even beneficial in passage 1 survive passage 2? Does this suggest that results from passage 1 are affected by complementation or interference in a major way? Also, why only show the first third of the genome?

To account for trans-complementation of the capsid that can occur in Passage 1 and we therefore showed P2 data exclusively for these genomic regions in the previous draft. In our revised manuscript, to avoid trans-complementation globally and for more consistent interpretation and presentation, we only show Passage 2 sequencing time points for all positions. We have modified all the figures to show the complete genomes for insertion, deletion, and substitution scanning experiments.

It would be more helpful to plot a quantity like 'gap density' along the EV A71 reference genome and integrate it into Fig 2A. Fig 2B is currently hard to compare with 2A because it is slightly stretched relative to the reference genome.

Thank you for this suggestion. We have revised this figure significantly to capture a more detailed description of InDel diversity across EV-A. This includes gap density, and a representation of Gap and deletion positions across a range of EV-A infecting both humans and non-human primates. We also measured Shannon Entropy across 482 complete EV-A71 sequences and plotted it alongside the effects of amino acid changes for a comparison of tolerance in our screen and in nature.

Distribution of fitness effects: One of the main results of this work is a quantification of what fraction of InDels are tolerated, are deleterious, or beneficial. Somewhat unsurprisingly, the great majority of such perturbations are lethal. What is more surprising, is how often a InDel is classified as beneficial. In fact, almost no InDels are neutral, while in some regions 10 or even 20% are beneficial. This makes me wonder how reliable this classification of non-lethal mutations into beneficial, neutral, and deleterious is. Did the authors quantify the accuracy of these fitness assignments in competition of individual mutants with the wildtype (or maybe single genotype growth curves to rule out complementation)? Such validation would increase the confidence in these results substantially.

We thank the reviewer for this insightful comment and apologize for the confusion in the terms used. Our analysis mainly aimed to classify variants according to their tolerance to InDels, and we readily admit that these most tolerated sites are in general much less fit than WT, now made clear through our competition assay reported in Supp. Fig 7.

We have already shown through deep sequencing of Enterovirus populations that InDels are rarely beneficial (<https://www.pnas.org/doi/full/10.1073/pnas.2304667120>). To clarify, we have modified our manuscript to calculate enrichment scores using a well-known software in the Deep Mutational Scanning field called Enrich2, and we have binned our variants according to low/medium/high tolerance. Our data were highly reproducible between the different biological replicates (See Supp. Figure 5).

Furthermore, to calculate the fitness effects of InDels or amino acid substitutions relative to wild type, we generated a new mutational scanning library containing all possible mutations (insertions, deletions, and amino acid changes) within a region covering the N-termini of VP1 and of 2A, to capture variants in both structural and non-structural regions (Supp. Figure 7 A, B). This experiment confirmed that, in general, most InDels were deleterious compared to wild type and amino acid changes (Supp. Figure 7 C). At best, the most tolerant InDels detected in our complete dataset were neutral or slightly beneficial compared to wild type/amino acid changes, more consistent with previous studies and your intuition (See Supp. Figure 7 D-H). The wild type normalized scores, and the previous measurements from the complete EV-A71 proteome showed a good linear relationship (Supp. Figure 7 I-O). So, we used a linear model for wild type normalization of our complete mutational scanning datasets, in 2M we connect these enrich2 scores show the WT-adjusted relative fitness for all three mutational scanning approaches.

In the section on phylogenetic analysis, I am not sure the statement "...allowed us to understand the emergence of InDels across EV-A species evolution" is warranted. You have described length variation among EV-A genomes and these results are consistent with the DMS results. But these results are descriptive rather than explanatory.

We have removed that statement.

Is it known whether they InDels in the termini of VP1 affect cleavage between VP1 and VP3 and 2A? I remember discussion about the ambiguous cleave site locations between VP3 and VP1 in EV-D 68 and if similar variation exists for EV-A71, it might affect the structural interpretation of the insertions discussed here.

We are unaware of any such ambiguity in the cleavage sites in EV-A71. The sites of tolerance in the VP1 N-term appear to fall downstream of conserved and functionally important structural features that have been shown to be critical to viral assembly and entry, suggesting that the observed tolerance is a feature of the extended VP1 N-terminus that is externalized during entry.

Minor suggestions:

Introduction: I felt the introduction a bit too black and white in the way it is contrasting nonsynonymous mutations and indels. Almost every major SC2 variant was characterized by specific indels in the N terminal domain of spike. Likewise, HIV-1 has frequent indel variation in its variable loops, sometimes even compensated frameshifts.

Thank you for this feedback, we agree that our statement about the evolutionary rate was misleading, and have clarified that these classes of mutation occur at different rates, but selection can drive these to rapidly accumulate.

Fig 2: the bar graph is hard to read. The majority of points are not visible since they correspond to lethal mutations. The entire deleterious range is compressed into the narrow 0 to 1 interval. I am wondering whether plotting these data on a logarithmic scale with a separate bin for lethal would do the data more justice. The same applied to Fig 2C&D: almost none of the relevant data can be seen. Here, I'd recommend a cumulative histogram with logarithmic axes.

We have improved the bar graphs and the histograms in Figure 2, with a zoomed-in panel showing Enrich2 scores that are above 0 to help with the visualization of the data.

Richard Neher

Responses to Reviewers

We thank the reviewers for their constructive feedback that has made the manuscript a much better final product. Sincere thanks for your time. - Patrick Dolan

Reviewer #1 (Remarks to the Author):

Bakhache et al. provide a well-written and detailed revision of their manuscript. One substantial change in this version is the removal of all structure prediction data due to different recommendation of reviewer 2 and myself on the general topic. To evaluate this decision at least two different use cases need to be distinguished for which structural predictions would be useful:

1) As a "scaffold" to map regions of higher InDel tolerance onto proteins with no deposited experimental structure and determine in which secondary structural elements InDels and substitutions are better tolerated as done in Fig. 4, 5, and S8.

From my viewpoint, there are no objections in the field right now to using structural predictions in the absence of experimental data for this kind of question as long as the models are well-validated along their scores (pLDDT, PAE, and pTM). An already 1,5 years old community assessment states: "In line with the reported high accuracy of the models, we found that AF2-predicted structures, on average, tend to give results that are as good as those derived from experimental structures."

(<https://www.nature.com/articles/s41594-022-00849-w>). I also respectfully disagree with reviewer 2's conclusion about AlphaFold 2 showing low accuracy in unstructured regions, in case understood his point correctly. Intrinsically disordered regions should have low localization accuracy by definition. Moreover, the presence of IDRs is generally predicted extremely well by AF2. So well that protein disorder databases such as MobiDB rely on AF2 to detect IDRs by their low pLDDT. Also, if a secondary structure has a high pLDDT in a generally well-folded prediction, I do not see any issue in calling this feature to evaluate InDel frequency.

Based on this, I do not see any reason not to use structural predictions for the remaining viral proteins as long as the scores are provided. In my opinion, including them, at least as a reference for future work, would be very useful.

2) To predict the effect of changes in regards to protein stability. Reviewer 2 and the authors are correct that this is a point of some debate in the field. However there is mounting evidence despite some negative reports that AF2 and now AF3 in combination with other methods such as "physics-based" modelling do work in evaluating the effect of sequence changes on protein stability and folding. However, I agree with the authors point that this might be out of the scope of this manuscript.

In conclusion, the authors have put a lot of extra work and effort into analysing the available experimental structures which makes the manuscript much stronger. I would appreciate if the authors would include AF2 or AF3 predictions of the remaining proteins for which no experimental structures are available, also to complete the analysis and

provide a reference for future use.

Otherwise I do not have any further objections and would like to congratulate the authors on this very well-designed study and the very well-written paper.

Thank you for your thoughtful comments. We agree with your points and have included AlphaFold3 predictions of the remaining structures NS2B and 3A in Supplemental Figure 9. We are excited to explore these tools more extensively in the future.

Reviewer #1 (Remarks on code availability):

I downloaded the Dryad depository and explored some of the code. Many of the scripts are well-documented, and the readme file gives a detailed but concise overview of the included data, scripts, and needed dependencies. However, due to time restrictions, I have not run any of the code.

Reviewer #2 (Remarks to the Author):

Fantastic job revising the manuscript. I appreciate all the changes the authors make and believe it to be an impactful manuscript with broad interest!

Willow Coyote-Maestas

Thank you, Willow, for your helpful insights.

Reviewer #3 (Remarks to the Author):

The authors have submitted an extensive revision of their manuscript and that address most points raised in my previous review. I think high throughput methodologies to assess the phenotypic impact of indel variation are important and in this light, this manuscript makes an important contribution. It is nice to see the correspondence between those locations with indel diversity among difference EV-A viruses.

Thank you, Richard. We agree the new figures showing diversity across hosts were very helpful. We appreciate your encouragement to develop a more compelling representation of the data.

remaining comment:

- the correlation of replicate results for indels in supp fig 5 D&E have some strange off-diagonal lines. Furthermore, why is the correlation more or less exact for some parts of the graphs for InDels, while one gets a more expected scatter for the substitutions?

We think these off diagonal lines and reflect the limitations of enrich2 to accurately capture fitness effects when variant counts are low. Recent work has put forward improve statistical methods to address this in substitution data,

e.g. 'ROSACE', but we have not implemented those recently reported methods here.

Response to reviewers

Reviewers Comments:

Reviewer #1 (Remarks to the Author):

I thank the authors for adding the AF3 predictions to the manuscript. I have no further queries and would like to congratulate the authors again on this very thoughtful study.

Thank you, again.

Reviewer #4 (Remarks to the Author):

Summary:

This work performed InDel mutagenesis of Enterovirus A71 to complement the current literature based on point mutations to understand protein tolerance. They generated libraries with an 8 amino acid insertion or 1, 2, or 3 amino acid deletion. Sequencing of the replication competent viruses identified tolerance hotspots, predominantly in 2A(pro) and VP1. Phylogenetic analysis was able to find evidence for the VP1 hotspot in the natural history of EV-A. This manuscript is clearly written and easy to follow.

Thank you for taking the time to review our work and bringing the issues below to our attention.

Your comments motivated us to make some significant improvements in how we were sharing the code, and we hope we have addressed the comments to your satisfaction. In the interest of facilitating reuse and further development of the scripts we have used herein, we have established two separate GitHub repositories:

- https://github.com/QVEU/InDel_Toolkit, including the python scripts that would be broadly useful for similar data sets, along with full installation and example datasets and command line instructions
- https://github.com/QVEU/eva71_dimple, containing all the scripts specific to the manuscript.

This will be available in addition to the current version of the scripts used to generate the data set, and all the associated data in the Dryad repository, previously made available: <https://doi.org/10.5061/dryad.866t1g1xm>

Minor:

VP1 is a high copy number capsid protein and has been used in the past as a candidate for the incorporation of tags. How does the hotspot identified in this work compare to the positions used in the literature? Does the screen identify a better position for tagging? See Stirk et al.

<https://doi.org/10.1093/protein/7.1.47> and Wang et al. <https://doi.org/10.1099/vir.0.040253-0>.

Thank you for bringing these to our attention, we have incorporated the references.

Reviewer #4 (Remarks on code availability):

Summary:

The authors provide numerous data files including their python and R code, and they deposited the sequencing runs at NCBI. The included readme files list the directory tree as well as descriptions of the included files.

However, I was unable to find demo data to test the code with, there are no running instructions, and the python dependencies were not listed. Demo data would not only facilitate the review process but also be helpful as a positive test of the pipeline for other labs that are interested in this technique.

We agree that the organization and utility of the code as it is housed at Dryad/Zenodo is not ideal for running, or reusing on new analyses. Therefore we have created a new Github repository, along with documentation that should clarify use of the python scripts.

Major

1. Demo data should be included with the python and R code. The description of the python scripts states that the input for both is a mapped .sam file. A small .sam file for testing in a reasonable amount of time should be included, along with a test file for the “query” and “templateFasta” arguments. It is unclear what the input files for the R scripts are.

We have now included demo data, example commands, and extensive documentation for the python scripts at the GitHub https://github.com/QVEU/InDel_Toolkit. You will find the required files, prerequisites, along with example command line usage for the demonstration data we have incorporated.

The Rscripts visualize the processed data for the manuscript. If you have a specific request with respect to demo data for the R scripts, please advise.

2. The authors should include running instructions. The R code returns numerous “No such file or directory” errors. Instructions should state what files the code is dependent on and the directory tree, e.g. should all of the files be in a single folder and the R working directory should be set to that folder.

We thank the reviewer for this, we suspect we did not have the code pointing to the correct relative path, this has been fixed, and we have confirmed it should run as expected now. We have also included the list of dependencies.

The instructions should also state that the python code should be run from the command line and include the command syntax. While one can deduce that from reading the .py file, it would be helpful if it was explicitly stated in the instructions with the arguments for calling the demo data. The instructions should also state what the input files are and what the output files should look like.

We have now included demo data, along with example that can be directly run after cloning the GitHub repository.

3. The dependencies/packages should be completely listed. For python, the dependencies should be listed and include the version numbers.

We have now included dependencies along with the versions we have tested. We expect it should be broadly compatible (except requiring python3)

This is done, please see the GitHub for InDel_Toolkit:

https://github.com/QVEU/InDel_Toolkit/blob/main/README.md

The R packages for Data_Generation_Markdown_InDel_Manuscript.r are listed in the readme files, but the packages for EVA71_EntropyAnalysis.R are missing.

We have now included dependencies.

Minor

1. There are 2 readme files: Dryad_Repo_InDel_Readme.txt and README.md. These appear to be mostly redundant and should be consolidated. I like the directory tree in README.md and found it helpful.

Unfortunately, this was something that was requested by the reviewers of our dryad repository. They've asked for very detailed accounting of the data in a specific structure through multiple rounds of peer review, so we would prefer they remain as is. We hope the github will clarify how to use these tools, and allow us and others to develop and improve them further.

2. In stickleback.0.2.py, should line 43 be <4? There are 3 arguments (pathto/input.sam, query, templateFasta) and python counts the file.pyargument as arg[0]. Therefore, if the right number of arguments are entered, it should be 4 (stickleback.py, pathto/input.sam, query, templateFasta).

if len(args)<4:

instead of

if len(args)<3:

Good catch, we should account for that. We had simply meant to account for cases where the user puts in no commands or something like "help"

Entering the command:

```
python .\stickleback.0.2.py input.sam query
```

Lead to the following error instead of the check that would state the usage:

```
template=args[3]
```

```
IndexError: list index out of range
```

- Please see the new github directory, with clarified readme and installation instructions.

We expect you'll have no problem with the example code.

https://github.com/QVEU/InDel_Toolkit/tree/main

3. stickleback.0.2.py imports "re" but doesn't use it.

- A vestigial import from an earlier version of the script.

4. DelMapper_v0.2.py imports “pandas” but doesn’t use it.

- Again, a vestigial import from an earlier version. Thanks for your careful reading of the code.

5. In DelMapper_v0.2.py, it looks like each function has a description commented above it. However, the function “parselist” has “posParse” written in the comment and the comment is the same for all 3 functions.

- Thank you, we apologize for this oversight and have now corrected the headers on the functions in the newest version of the script.

6. It would be nice if DelMapper_v0.2.py also included a syntax check and return the usage when an incorrect number of arguments are used, like stickleback.0.2.py.

- Thank you, we have incorporated these changes. See the updated code in the Github repository.

4th September 2024

Response to reviewers:

We appreciate the time and effort of the reviewers. We have responded to specific points below.

Reviewer Expertise:

Referee #4: Code

Referee #5: Deep mutational scanning

Reviewers Comments:

Reviewer #4 (Remarks to the Author):

The github pages are very well organized with clear instructions.

- **Thank you.**

Testing:

stickleback.py: runs as explained

- **Thank you.**

smelt.py example and usage are incorrect

The example command is missing the template.fasta argument.

The usage is missing the output.csv argument.

- **We fixed this mistake in the example readme on the GitHub (https://github.com/QVEU/InDel_Toolkit).**

In addition, it requires the outdated numpy v1.x

- **We disagree that pre-2.0 numpy is outdated (it certainly was not before the release of numpy 2.0 in June 2024, six months after the submission of this manuscript. It's well known that numpy 2+ breaks many other dependencies ('breaking changes') and we would not consider it stable. We've noted in the GitHub which version was used in the analysis and tested for compatibility.**

EVA71_EntropyAnalysis.R: runs as explained

- **OK, thank you.**

DMS_Processing_Workflow.R: runs as explained in RStudio but not R.

In R, every call to the `codonFilter()` function generates the following error:

```
> capsid_P0_filtered <- codonFilter(oligos_programmed_capsid,
"input_files/codon_counts/Capsid_input_DMS.codonCounts", wildtype_capsid, start_end_capsid)
Rows: 81889 Columns: 1
— Column specification—
Delimiter: "\t"
chr (1): >Capsid_DMS-1_Met2Cys

i Use `spec()` to retrieve the full column specification for this data.
i Specify the column types or set `show_col_types = FALSE` to quiet this message.
Rows: 2194 Columns: 67
— Column specification—
Delimiter: "\t"
dbl (67): AAA, AAC, AAG, AAT, ACA, ACC, ACG, ACT, AGA, AGC, AGG, AGT, ATA, ATC, ATG, ATT, CAA,
CAC, CAG, CAT, CCA, CCC, CCG, CCT, CGA...

i Use `spec()` to retrieve the full column specification for this data.
i Specify the column types or set `show_col_types = FALSE` to quiet this message.
Error in h(simpleError(msg, call)) :
error in evaluating the argument 'x' in selecting a method for function 'slice': Rle of type
'list' is not supported
```

- **We have noted that we expect users to run this interactively in Rstudio, or in a dedicated conda environment with tidyverse and r installed to avoid any potential interactions with other install r packages (perhaps, for example, `S4::slice`). (see the github: https://github.com/QVEU/eva71_dimple). However, we have not been able to reproduce this reviewer's 'slice()' error running the following in the command line, and the code completed as expected with no issues. Without more information about how the reviewer ran the code, we can't diagnose their problem. The other text shared is just verbose warnings from `tibble()`, and is not an issue.**

This should work for all users with conda installed, and we've included in the newly updated GitHub:

```
```\nconda create -n newenv\nconda activate newenv\nconda install -c conda-forge r --solver=classic\nconda install -c conda-forge r-tidyverse --solver=classic
```

Rscript DMS\_Processing\_Workflow.R

'''

### **Running this worked for us on multiple machines.**

Major:

1. Please state the version of the python dependencies. As outlined above, this is important for smelt.py
  - **Done.**
2. Correct the smelt.py example and usage information. Make it clear that it uses an outdated numpy version or update the code to use the current version of numpy.
  - **See above.**
3. Check if you can reproduce the DMS\_Processing\_Workflow.R error in a fresh install of R. If so, please make a comment on github.
  - **See above.**

Reviewer #5 (Remarks to the Author):

Overall, this is a good study. It provides a wealth of information enterovirus A, and also is the most comprehensive study to date on insertions versus deletions versus substitutions to a viral protein.

- **Thank you for your time and effort reviewing. We appreciate your kind words.**

Minor comments:

- In Figure 1A-F, one surprising aspect is that some deletions and insertions (as well as substitutions) have extremely positive scores, suggesting they improve growth a lot. Is this a real result or experimental noise? Ideally this could be validated for some key mutations, or at least discussed. -

- **The reviewer is correct that the enrichment values for some variants are quite positive. However, this doesn't equate to fitness as we discussed with reviewers in the first few round of review. Please see Figure 1M (added in our first revision) in the reviewed version of the manuscript, where we performed a pooled competition experiment to rescale these measurements relative to the wild type control, better reflecting 'fitness' rather than enrichment. The reviewer will notice that these fitness values**

**are actually quite modest, but relative to other insertions or deletions, which are largely lethal to virus viability, have very large enrichment values.**

- In the text, a bit more discussion should be given to the fact that the insertional mutants are all of a specific length and context (glycine-serine flanked). It is possible that the effects of insertions would be different if they were shorter, etc. I don't expect the authors to test every possible insertion length/type (which are basically infinite), but they should more clearly explain they are testing a specific length and context.

- **We have added an extended discussion to lines 88, and 137-138.**
  - ***88: One caveat of this approach is that the BsaI sites used for the incorporation of the insert remain, leaving a fixed sequence flanking the inserted sequence of interest.***

- Line 47-48: A few recent viral protein DMS experiments have included some deletions. Those studies don't have insertions and are not nearly as comprehensive as this one, so they do not in any way decrease the novelty and impact of the current study. However, for completeness they should be cited, eg: deletions in SARS-CoV-2 RBD (<https://journals.plos.org/plospathogens/article?id=10.1371/journal.ppat.1011901>), deletions in some sites in SARS-CoV-2 spike including NTD (<https://www.nature.com/articles/s41586-024-07636-1>)

- **Thank you, we have added these references to our discussion of tolerance to substitution and deletion. Highlighting the emerging observations that enveloped and non-enveloped viruses exhibit very different tolerance profiles. See below:**

**Disc. lls. 234-247 (new text in italics):**

**One key observation emerging from our study is the complex constraints governing InDel and substitution tolerance in the EV-A71 capsid. The capsid proteins thread a difficult evolutionary needle, maintaining an extracellular and metastable state that protects the genome in harsh conditions while also being labile enough to rapidly uncoat after engaging host receptors or host environments. They are also the target of selection, usually carrying key immune epitopes. In our screen, VP1 had the greatest concentration of positions with high relative tolerance to all mutation classes, specifically at the N and C termini (Figure 4 U, V), consistent with our analysis of natural variation across EV-A (Figures 1L, 5A and 5D). *These host- and viral-facing functions overlap in picornaviral capsid proteins, however, in enveloped viruses such as flaviviruses and coronaviruses, these functions are largely partitioned between their spike or envelope proteins and their nucleocapsid proteins, which appears to relax***

***evolutionary constraints on these proteins10,42. Consistent with this, studies examining the effects of mutations, including single-codon deletions, in the Spike protein of CoVs revealed much greater tolerance to deletions than observed here, instead largely mirroring where substitutions were tolerated.43,44***